

# Infinite pseudo-conformal symmetries of classical

# $T\bar{T}$, $J\bar{T}$ and $JT_a$ - deformed CFTs

**Monica Guica[1,2,3] and Ruben Monten[1,4]**

**1** Institut de Physique Théorique, CEA Saclay, CNRS, 91191 Gif-sur-Yvette, France
**2** Department of Physics, Stockholm University, AlbaNova, 106 91 Stockholm, Sweden
**3** Nordita, Roslagstullsbacken 23, SE-106 91 Stockholm, Sweden
**4** Mani L. Bhaumik Institute for Theoretical Physics, Department of Physics & Astronomy,
University of California, Los Angeles, CA 90095, USA

## Abstract

We show that $T\bar{T}$, $J\bar{T}$ and $JT_a$ - deformed classical CFTs posses an infinite set of symmetries that take the form of a field-dependent generalization of two-dimensional conformal transformations. If, in addition, the seed CFTs possess an affine $U(1)$ symmetry, we show that it also survives in the deformed theories, again in a field-dependent form. These symmetries can be understood as the infinitely-extended conformal and $U(1)$ symmetries of the underlying two-dimensional CFT, seen through the prism of the "dynamical coordinates" that characterise each of these deformations. We also compute the Poisson bracket algebra of the associated conserved charges, using the Hamiltonian formalism. In the case of the $J\bar{T}$ and $JT_a$ deformations, we find two copies of a functional Witt - Kac-Moody algebra. In the case of the $T\bar{T}$ deformation, we show that it is also possible to obtain two commuting copies of the Witt algebra.

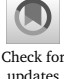

## 1. Introduction and summary

Two-dimensional quantum field theories universally contain a composite irrelevant operator denoted as $T\bar{T}$, with a number of remarkable properties [1]. When added to the action, this operator generates a deformation [2,3] that alters the UV drastically and non-locally, but in a controllable way [4–6]. Quantities such as the finite-volume energy spectrum, the S-matrix and various partition functions can be computed exactly in terms of their undeformed counterparts [2–12]. At the classical level, the deformed Hamiltonian can be written down exactly in terms of the undeformed one [13], while at the quantum level it is possible, to a certain extent, to track the flow of energy eigenstates and of correlation functions of distinguished operators [14–16].

In addition to their appeal as solvable non-local field theories, $T\bar{T}$ - deformed QFTs have found interesting applications in various fields, ranging from QCD [17–21] and string theory [22–24] to holography[1] [26–28]. Various properties and generalizations were explored in [29–47]. A particularly interesting off-shoot is a single-trace analogue of the $T\bar{T}$ deformation, proposed in [48] and further developed in [49–54], which opens up a path towards holography for asymptotically flat spacetimes in terms of QFTs that share the ultraviolet properties of $T\bar{T}$.

As discussed in [2], QFTs with similar properties to $T\bar{T}$ can be obtained by considering deformations constructed from the components of any two conserved currents. A particular case that has been studied rather thoroughly is the $J\bar{T}$ deformation [55], constructed from the components of a $U(1)$ current and those of the right-moving stress tensor, as well as its generalization that we will denote as $JT_a$, where the right-moving stress tensor is replaced by the generator $T_{\mu a}$ of translations in some fixed direction $\hat{x}^a$ [39,56,57]. The spectra of the deformed theories can again be worked out exactly [56,58–60] and the UV is again modified in a non-local, yet predictable way. While certain physical properties of $J\bar{T}$ and $JT_a$ - deformed QFTs are quite different from those of $T\bar{T}$ - deformed ones (for example, these deformations break Lorentz invariance), they still share many important properties with it and can offer a complementary point of view. Indeed, the $J\bar{T}$ and $JT_a$ deformations are simpler than $T\bar{T}$ in the sense that they are only non-local along one direction; in particular, $J\bar{T}$ - deformed CFTs

---

[1]See also the lecture notes at [25] for a pedagogical discussion of the existing holographic proposals for $T\bar{T}$ - deformed CFTs.

possess full one-dimensional conformal invariance, allowing for the calculation of correlation functions [61]. Additionally, the single-trace analogue of $J\bar{T}$ - deformed CFTs [58, 59, 62] is a promising avenue for understanding holography for extremal black holes [63] in terms of QFTs that share the asymptotic UV behaviour of $J\bar{T}$ - deformed CFTs.

In view of these applications and possible generalizations, it is of interest to better understand the structure of $T\bar{T}, J\bar{T}$ and $JT_a$ - deformed QFTs. A very basic question is what their symmetries are, and in particular whether these symmetries can be infinitely extended, as is common in two-dimensional QFTs. For example, it has already been shown that all deformations of the Smirnov–Zamolodchikov type preserve integrability, if initially present [2], which amounts to an infinite number of conserved charges.

In this article, we study the $T\bar{T}$, $J\bar{T}$ and $JT_a$ deformation of *classical* two-dimensional CFTs. As is well known, the seed theories enjoy an infinite-dimensional symmetry that consists (at the classical level), of two commuting copies of the Witt algebra that are enhanced, if additional $U(1)$ symmetries are present, to two commuting copies of the Witt - Kac-Moody algebra. Our task is to determine whether these symmetries are preserved by the various deformations.

The holographic analyses of [28] for the case of $T\bar{T}$ and [64] for the case of $J\bar{T}$ suggest that the answer is affirmative. These papers studied the asymptotic symmetries of AdS$_3$ associated to the mixed boundary conditions induced by each of these deformations, and found that all the Virasoro and Kac-Moody symmetries are still present, though their generators become non-local due to their dependence on a certain field-dependent modification of the coordinates.

While these holographic analyses are indicative, they remain an indirect and purely classical (large $N$) calculation, despite giving access to the central extension of the symmetry algebra. In this article, we perform a different classical analysis of the symmetries, this time from a field-theoretical point of view. Even though the central extensions are no longer accessible — and thus it will only be sensitive to the Witt part of the Virasoro algebra — this calculation has the advantage of being more straightforwardly generalizable to the quantum case, which is the question of true interest. We will show that an infinite-dimensional set of symmetries is indeed preserved, and we exemplify how the Poisson brackets of their conserved charges furnish Witt and Kac-Moody algebras in the deformed theory. If this structure survives at quantum level, it would represent a non-local generalization of two-dimensional CFTs, with a possibly generalized symmetry structure that is yet to be understood.

The main reason to expect the original symmetries to remain present is that, as nicely explained in [65] (see also [4,66]), the main effect of the $T\bar{T}$ deformation — and, as it turns out, of all the other deformations in the Smirnov–Zamolodchikov class — is to provide a dynamical set of coordinates, through which the underlying QFT dynamics is seen. This was shown in [66] at the level of the classical equations of motion, while [13] (see also [56]) studied it in Hamiltonian language, showing the deformed and undeformed systems are canonically equivalent. Therefore, from the perspective of the dynamical coordinates, it can be expected that all the rigid CFT structure should be present, even if deformed in a predictable way.

The logic of this paper is a very simple generalization of the argument that the symmetry group of two-dimensional CFTs is infinite-dimensional. We consider an arbitrary translationally-invariant field theory on flat space, with null coordinates $U, V = \sigma \pm t$, and we perform the coordinate shift

$$U \to U + \epsilon f(u), \quad V \to V - \epsilon \bar{f}(v), \tag{1.1}$$

where $u(U,V), v(U,V)$ are some possibly field-dependent "coordinates" and the functions $f(u), \bar{f}(v)$ are *arbitrary*. Concentrating on the left-movers for simplicity, the change in the

action under the shift of $U$ takes the form

$$\delta_f S = -\int dU dV \, \epsilon f'(u) \left( T_{UU} \partial_V u + T_{VU} \partial_U u \right).$$ (1.2)

If the QFT is question is a CFT, then the trace of the stress tensor, $T_{VU}$, vanishes off-shell. By taking $u = U$, the variation of the action vanishes for any function $f(U)$ yielding, upon inclusion of the right-movers, the well-known infinite-dimensional conformal symmetries of two-dimensional CFTs.

In the case of $T\bar{T}$ , $J\bar{T}$ and $JT_a$ - deformed CFTs, the trace of the stress tensor no longer vanishes, but it is still true that the stress tensor only has two independent components off-shell. To have an infinite-dimensional symmetry, we simply choose the function $u(U,V)$ so that (1.2) is zero. The solution for $u$ will be field-dependent, yet its structure will turn out to be universal for each particular type of deformation, due to their very special nature. For example, the field-dependent coordinates $u^a = \{u, v\}$ for $JT_a$-deformed CFTs (including the special case of $J\bar{T}$), are simply

$$u^a = U^a - \lambda^a \varphi,$$ (1.3)

where $\lambda^a$ is the deformation *vector* and $\varphi$ is the scalar dual to the current $J$ via $J = \star d\varphi$. In $T\bar{T}$ - deformed CFTs, the off-shell solution for $u$ is harder to write explicitly. On-shell, it can be chosen to agree with the expressions that previously appeared in the literature [4].

The field-dependent coordinates $u, v$ obtained this way are the same as those in which the dynamics of the deformed CFT is mapped to that of the original CFT [4]. Therefore, in a sense, the infinite-dimensional symmetries we are uncovering are nothing but the infinite conformal symmetries of the original CFT, seen through the prism of the dynamical coordinates. For this reason, we call these symmetries pseudo-conformal.

While the concept of a field-dependent spacetime transformation may seem somewhat unusual, it is definitely not new [67,68]. To make it more palatable, note that it can be simply understood as a neat way of packaging an infinite set of rather standard non-linear and field-dependent transformation laws for each of the fields in the theory, as well as all derivatives of these fields. These non-linear transformations are symmetries because they leave the action invariant.

When the seed CFT also contains conserved currents (taken to be $U(1)$ for simplicity), it is well known that they generate an infinite-dimensional $U(1)$ Kac-Moody symmetry. To see this, consider the current $\chi(u)J_A$ associated with a phase rotation of the fundamental fields parametrized by an arbitrary function $\chi(u)$ of some possibly field-dependent coordinate $u$, where $J_A$ is conserved. The divergence of this new current is

$$\nabla^A(\chi(u)J_A) = 2\chi'(u)(J_U \partial_V u + J_V \partial_U u).$$ (1.4)

In a CFT, it is usually possible to choose a linear combination of the $U(1)$ current generating the global phase shift and its topological counterpart so that the $V$ component of the sum vanishes. Then, $\chi(U)J_A$ is conserved for arbitrary transformations $\chi(U)$, giving rise to an infinite-dimensional Kac-Moody symmetry. In the $T\bar{T}, J\bar{T}$ and $JT_a$ - deformed CFTs, it will generally no longer be possible to find such chiral linear combinations of the currents; however, we will show that in each of these cases, the deformed structure does allow us to find linear combinations of the conserved currents such that the ratio of their components is the same as that of the stress tensor components, implying that (1.4) will vanish for the *same* function $u(U,V)$ as (1.2). An analogous statement holds for the right-movers. Consequently, we will find that the original $U(1)$ affine symmetries of the CFT get deformed into an infinite-dimensional set of field-dependent $U(1)$ symmetries, which involve the same field-dependent coordinates as the pseudo-conformal ones.

Having established the existence of an infinite set of field-dependent symmetries, a natural follow-up question is to find their algebra, and in particular whether it coincides with that of the seed CFT. The holographic analyses of [28, 64] suggest that the answer is yes, based on a calculation that heavily relies on the on-shell map to a CFT living in the field-dependent coordinates. In anticipation of the quantum case, where these coordinate transformations become operator-valued and thus significantly more difficult to work with, we decided to employ the Hamiltonian formalism to compute the Poisson brackets of the symmetry generators.

The Hamiltonian treatment for $T\bar{T}$ - deformed CFTs was developed in [13] (see also [56]), who gave a very simple expression for the deformed Hamiltonian *density* in terms of the undeformed Hamiltonian and momentum densities. We will show that this method can be easily adapted to the $J\bar{T}$ and $JT_a$ deformations to obtain the corresponding expression for the deformed Hamiltonian density in terms of the undeformed Hamiltonian, momentum and charge densities. This allows us to compute the algebra of the conserved charges in the deformed theories using the universal equal-time Poisson brackets of the various currents in the undeformed CFT. For the simpler $J\bar{T}$ and $JT_a$ deformations, the algebra can readily be shown to consist of two commuting copies of a functional Witt - Kac-Moody algebra. The $T\bar{T}$ calculation is slightly more involved, as we encounter certain ambiguities in the Poisson bracket of the field-dependent coordinates. However, we show it is possible to fix this freedom in such a way that the algebra consists again of two commuting copies of the functional Witt algebra. Throughout the paper, we concentrate on the charge algebra on compact space, as it presents several subtleties that disappear when taking the flat space limit.

The plan of this paper is as follows. In section 2, we exhibit the infinite-dimensional symmetries of $T\bar{T}$ - deformed CFTs, compute the Poisson bracket algebra of the conserved charges and show that it can be chosen to be Witt. We conclude with a possibly instructive comparison of the Lagrangian and Hamiltonian treatments of the $T\bar{T}$ - deformed free boson. In section 3, we discuss the $J\bar{T}$ deformation, which provides a nice warm-up for the more general $JT_a$ case. We start by exhibiting the infinite symmetries of a $J\bar{T}$ - deformed free boson, then we compute the charge algebra, and finally show that the results generalize to arbitrary $J\bar{T}$ - deformed CFTs, by providing a general Hamiltonian formulation of these theories along the lines of [13]. In section 4, we repeat the same exercise for the $JT_a$ deformation. Various details of the rather technical calculations are relegated to the appendices.

## 2. $T\bar{T}$ - deformed CFTs

We consider a family of classical, Lorentzian field theories parameterized by a real coupling $\mu$ of length dimension two, so that the theory at $\mu = 0$ is a CFT. Their actions are related by the following flow equation

$$\partial_\mu S_L(\mu) = \frac{1}{2} \int d^2x \sqrt{\gamma} (T^{\alpha\beta} T_{\alpha\beta} - T^2), \tag{2.1}$$

where the right-hand side is evaluated in the theory deformed by an amount $\mu$. This equation can be solved for the deformed Lagrangian; a well-known example is the $T\bar{T}$ - deformed free boson, which is related to the Nambu–Goto action in static gauge [3].

The trace of the stress tensor in a $T\bar{T}$ - deformed CFT no longer vanishes; however, it is still true that the stress tensor only has two independent components off-shell, due to the trace flow equation

$$T^\alpha{}_\alpha \equiv T = -\mu(T^{\alpha\beta} T_{\alpha\beta} - T^2), \tag{2.2}$$

which was proven in e.g. [28] for general classical CFTs.

One can equivalently perform a Hamiltonian analysis of the system [13], by solving the associated flow equation for the Hamiltonian density

$$\partial_\mu \mathcal{H} = -\frac{1}{2}(T^{\alpha\beta}T_{\alpha\beta} - T^2), \tag{2.3}$$

where the components of the stress tensor are obtained from the Hamiltonian density via[2]

$$T_{tt} = \mathcal{H}, \quad T_{\sigma t} = \partial_{\pi_k}\mathcal{H}\,\partial_{\phi'^k}\mathcal{H}, \quad T_{t\sigma} = \pi_k \phi'^k \equiv \mathcal{P}, \quad T_{\sigma\sigma} = \pi_k \partial_{\pi_k}\mathcal{H} + \phi'^k \partial_{\phi'^k}\mathcal{H} - \mathcal{H}, \tag{2.4}$$

with $\phi_k, \pi_k$ denoting the canonical fields and their conjugate momenta. The relation between the flow (2.1) of the Lagrangian and the above flow of the Hamiltonian has been explained in [16].

The symmetries of the seed theory impose certain constraints on the undeformed Hamiltonian $\mathcal{H}^{(0)}$. First, Lorentz invariance allows us to choose the stress tensor to be symmetric, so $T_{\sigma t}^{(0)} = T_{t\sigma}^{(0)} = \mathcal{P}$. Then, the conformal symmetry of the undeformed CFT guarantees that the stress tensor is traceless, $T_{\sigma\sigma}^{(0)} = T_{tt}^{(0)} = \mathcal{H}^{(0)}$. This requires $\mathcal{H}^{(0)}$ to satisfy

$$(\pi_k \partial_{\pi_k} + \phi'^k \partial_{\phi'^k})\mathcal{H}^{(0)} = 2\mathcal{H}^{(0)}. \tag{2.5}$$

By solving the flow equation, [13] showed that the stress tensor stays symmetric and that the deformed Hamiltonian density takes the simple form

$$\mathcal{H} = \frac{1}{2\mu}\sqrt{1 + 4\mu\mathcal{H}^{(0)} + 4\mu^2\mathcal{P}^2} - \frac{1}{2\mu}, \tag{2.6}$$

while $T_{t\sigma} = \mathcal{P}$ remains unchanged. Using (2.6) and (2.5), $T_{\sigma\sigma}$ can be expressed only in terms of $\mathcal{H}$ and $\mathcal{P}$, thus showing that the deformed stress tensor still only has two independent components, which algebraically determine the rest.

Note the equation (2.6) is similar to the corresponding equation for the energies of $T\bar{T}$-deformed theories on a compact space. However, eq. (2.6) is an equation for the Hamiltonian density - a local current - and not the integrated charge.

## 2.1. Infinite pseudo-conformal symmetries

We would now like to investigate the field-dependent pseudo-conformal symmetries of $T\bar{T}$-deformed CFTs. As explained in the introduction, we consider shifting the null coordinates $U, V = \sigma \pm t$ as

$$U \to U + \epsilon f(u), \quad V \to V - \epsilon\bar{f}(v) \tag{2.7}$$

for *arbitrary* functions $f(u), \bar{f}(v)$ of some possibly field-dependent coordinates $(u, v)$, which depend on both $U$ and $V$. The change in the action takes the form

$$\delta_f S = -\int dU dV \,\epsilon f'(u)\,(T_{UU}\partial_V u + T_{VU}\partial_U u),$$

$$\delta_{\bar{f}} S = \int dU dV \,\epsilon \bar{f}'(v)\,(T_{VV}\partial_U v + T_{UV}\partial_V v). \tag{2.8}$$

A convenient parametrization of the two independent stress tensor components is in terms of

$$\mu\mathcal{L} \equiv -\frac{T_{UV}}{T_{VV}} = \frac{2\mu T_{UU}}{1 - 2\mu T_{UV}}, \quad \mu\bar{\mathcal{L}} \equiv -\frac{T_{VU}}{T_{UU}} = \frac{2\mu T_{VV}}{1 - 2\mu T_{VU}}, \tag{2.9}$$

---

[2]Note that our conventions are related to those of [13] by $\mathcal{H} \to -\mathcal{H}, \mathcal{P} \to -\mathcal{P}$.

where we used the trace identity (2.2), which in null coordinates reduces to $T_{UV} = -2\mu(T_{UU}T_{VV} - T_{UV}^2)$, in the second step. In terms of $\mathcal{L}, \bar{\mathcal{L}}$, the components of the stress tensor are

$$T_{UU} = \frac{\mathcal{L}}{2(1-\mu^2\mathcal{L}\bar{\mathcal{L}})}, \qquad T_{VV} = \frac{\bar{\mathcal{L}}}{2(1-\mu^2\mathcal{L}\bar{\mathcal{L}})}, \qquad T_{UV} = -\frac{\mu\mathcal{L}\bar{\mathcal{L}}}{2(1-\mu^2\mathcal{L}\bar{\mathcal{L}})} \qquad (2.10)$$

and its conservation equations read

$$(\partial_V - \mu\bar{\mathcal{L}}\,\partial_U)\mathcal{L} = 0, \qquad (\partial_U - \mu\mathcal{L}\,\partial_V)\bar{\mathcal{L}} = 0. \qquad (2.11)$$

This parametrization is useful for identifying the symmetries of the $T\bar{T}$ - deformed CFTs. The variation (1.2) of the action will vanish for *arbitrary* $f(u), \bar{f}(v)$, provided the field-dependent coordinates $u, v$ are chosen to satisfy

$$\partial_V u = \mu\bar{\mathcal{L}}\,\partial_U u, \qquad \partial_U v = \mu\mathcal{L}\,\partial_V v. \qquad (2.12)$$

Assuming that $\mathcal{L}$ and $\bar{\mathcal{L}}$ are analytic functions of $(U, V)$, the Cauchy—Kowalevski theorem guarantees that a solution to these two partial differential equations exists, at least locally. Once the initial values are specified (for example on a hypersurface of constant $V$ and constant $U$), the solution is unique. Formally, we can write the solutions to (2.12) as

$$u = \mathcal{P}e^{\mu\int dV\bar{\mathcal{L}}\partial_U}u_0(U), \qquad v = \mathcal{P}e^{\mu\int dU\mathcal{L}\partial_V}v_0(V), \qquad (2.13)$$

for some functions $u_0(U), v_0(V)$, where $\mathcal{P}$ denotes path-ordering. Since the solution exists *off-shell* for any $\mathcal{L}, \bar{\mathcal{L}}$, (2.7) are symmetries of the classical action.

An interesting feature of the field-dependent coordinates is that the stress tensor conservation equations (2.11) simply reduce to (anti-)holomorphicity of $\mathcal{L}, \bar{\mathcal{L}}$ with respect to $u, v$

$$(\partial_V - \mu\bar{\mathcal{L}}\,\partial_U)\mathcal{L} = (\partial_V u - \mu\bar{\mathcal{L}}\partial_U u)\partial_u\mathcal{L} + (\partial_V v - \mu\bar{\mathcal{L}}\partial_U v)\partial_v\mathcal{L} = \partial_V v(1 - \mu^2\mathcal{L}\bar{\mathcal{L}})\partial_v\mathcal{L} = 0 \quad (2.14)$$

and similarly for $\bar{\mathcal{L}}$. Thus, in terms of $\mathcal{L}, \bar{\mathcal{L}}$ and the field-dependent coordinates $u, v$, the dynamics of the stress tensor in the $T\bar{T}$ - deformed CFT becomes that of a usual CFT stress tensor, where *on-shell* we have $\mathcal{L} = \mathcal{L}(u)$ and $\bar{\mathcal{L}} = \bar{\mathcal{L}}(v)$.

The expression (2.13) for the field-dependent coordinates is rather formal, and it would be good to find a more manageable one. To this end, we consider the inverse transformations $U(u, v), V(u, v)$. Using (2.12), they can be shown to satisfy

$$\partial_v U = -\mu\bar{\mathcal{L}}\partial_v V, \qquad \partial_u V = -\mu\mathcal{L}\partial_u U. \qquad (2.15)$$

These equations are very simple to solve on-shell, as the (anti-)holomorphicity of $\mathcal{L} = \mathcal{L}(u)$ and $\bar{\mathcal{L}} = \bar{\mathcal{L}}(v)$ implies that $\partial_u\partial_v U = \partial_u\partial_v V = 0$. Thus, on-shell $\partial_u U$ and $\partial_v V$ are (anti-)holomorphic functions of $u$ and $v$, respectively, that can be chosen at will. A simple choice is $\partial_u U = \partial_v V = 1$, which will allow us to recover the well-known field-dependent coordinate transformation[3] [4,66]

$$U = u - \mu\int^v dv'\bar{\mathcal{L}}(v'), \qquad V = v - \mu\int^u du'\mathcal{L}(u). \qquad (2.17)$$

---

[3]It is interesting to ask whether such an additional constraint on $U, V$ is also possible off-shell. The answer is no, as can be seen by taking derivatives of (2.15)

$$\Box U = \partial_u\partial_v U = \frac{-\mu\partial_u\bar{\mathcal{L}}\partial_v V + \mu^2\bar{\mathcal{L}}\partial_v\mathcal{L}\partial_u U}{1-\mu^2\mathcal{L}\bar{\mathcal{L}}}, \qquad \Box V = \partial_u\partial_v V = \frac{-\mu\partial_v\mathcal{L}\partial_u U + \mu^2\mathcal{L}\partial_u\bar{\mathcal{L}}\partial_v V}{1-\mu^2\mathcal{L}\bar{\mathcal{L}}}. \qquad (2.16)$$

If $\partial_u U = 1$, the left-hand side of this equation vanishes, whereas the right-hand side clearly does not, off-shell. An explicit off-shell solution to these equations can be found in the free boson case, discussed at the end of this section.

So far, we have shown that the action is invariant under the field-dependent coordinate transformations (2.7), with $u, v$ chosen to satisfy (2.12) off-shell. The conserved charges associated with these symmetries are given by the general formula

$$Q_\xi = \int d\sigma\, n^A T_{AB} \xi^B, \qquad (2.18)$$

where $n = \partial_t$ is the unit normal to a constant time slice and $\xi^A$ are the coordinate shifts in eq. (2.7), namely $\xi = f(u)\partial_U$ for the first symmetry and $\xi = -\bar{f}(v)\partial_V$ for the second. The explicit form of the charges is

$$Q_f = \int d\sigma\, f(u)(T_{UU} - T_{VU}) = \frac{1}{2}\int d\sigma\, f(u)\mathcal{L}\frac{1 + \mu\bar{\mathcal{L}}}{1 - \mu^2\mathcal{L}\bar{\mathcal{L}}} \qquad (2.19)$$

and

$$\bar{Q}_{\bar{f}} = \int d\sigma\, \bar{f}(v)(T_{VV} - T_{UV}) = \frac{1}{2}\int d\sigma\, \bar{f}(v)\frac{\bar{\mathcal{L}}(1 + \mu\mathcal{L})}{1 - \mu^2\mathcal{L}\bar{\mathcal{L}}}. \qquad (2.20)$$

The integrands can be written in a more suggestive form by noticing that[4]

$$u' \equiv \partial_\sigma u = \partial_U u(1 + \mu\bar{\mathcal{L}}) = \frac{1 + \mu\bar{\mathcal{L}}}{(1 - \mu^2\mathcal{L}\bar{\mathcal{L}})\partial_u U}, \qquad v' \equiv \partial_\sigma v = \frac{1 + \mu\mathcal{L}}{(1 - \mu^2\mathcal{L}\bar{\mathcal{L}})\partial_v V}. \qquad (2.22)$$

Using this, the expressions for the charges become

$$Q_f = \frac{1}{2}\int d\sigma\, \partial_\sigma u\, \partial_u U\, \mathcal{L} f(u), \qquad \bar{Q}_{\bar{f}} = \frac{1}{2}\int d\sigma\, \partial_\sigma v\, \partial_v V\, \bar{\mathcal{L}}\bar{f}(v). \qquad (2.23)$$

This rewriting makes charge conservation manifest, as on-shell one is integrating purely (anti)-holomorphic functions. It also shows that the pseudo-conformal conserved charges of $T\bar{T}$ - deformed CFTs precisely correspond to those of an auxiliary CFT whose coordinates are $u, v$, up to the rescaling factors $\partial_u U, \partial_v V$ which, as explained, can be chosen to equal one on-shell. These expressions then coincide with those found in the holographic analysis of [28].

Note that when the $T\bar{T}$ - deformed CFT is placed on a compact space, charge conservation requires $f(u)$ to be a periodic function of $u$, and similarly for $\bar{f}(v)$. Since the periodicity of the coordinates $U, V$ of the field theory is fixed, that of the solutions to (2.12) will in general be field-dependent. Denoting these field-dependent radii of $u, v$ as $R_u$ and, respectively, $R_v$, then the argument of $f(u)$ is more precisely $\hat{u} = u/R_u$, from which it is straightforward to build a periodic function, thus ensuring the conservation of the associated charge. The explicit dependence on the field-dependent radii will become important in the next section, as they contribute non-trivially to the Poisson brackets.

## 2.2. Conserved charges and Poisson bracket algebra

In this section, we proceed to computing the Poisson bracket algebra of the conserved charges (2.19) and (2.20) in a $T\bar{T}$ - deformed CFT placed on a cylinder of radius $R$. The result for non-compact space can be simply obtained by taking $R \to \infty$. One reason we concentrate on the compact case is that, as we will see, it presents a number of subtleties that are not present in the non-compact setting.

---

[4]The map between the matrices $\partial_X x$ and $\partial_x X$ is explicitly given by

$$\begin{pmatrix} \partial_u U & \partial_v U \\ \partial_u V & \partial_v V \end{pmatrix} = \frac{1}{\partial_u u\, \partial_v v(1 - \mu^2\mathcal{L}\bar{\mathcal{L}})}\begin{pmatrix} \partial_v v & -\mu\bar{\mathcal{L}}\partial_U u \\ -\mu\mathcal{L}\partial_V v & \partial_U u \end{pmatrix}. \qquad (2.21)$$

We start by rewriting the charges in the Hamiltonian language. Using (2.4), we find the simple expressions

$$Q_f = \frac{1}{2}\int_0^R d\sigma f(\hat{u})(\mathcal{H}+\mathcal{P}), \qquad \bar{Q}_{\bar{f}} = \frac{1}{2}\int_0^R d\sigma \bar{f}(\hat{v})(\mathcal{H}-\mathcal{P}), \qquad (2.24)$$

where $\hat{u} = u/R_u$ and $\hat{v} = v/R_v$. The relation between $\mathcal{L}, \bar{\mathcal{L}}$ and $\mathcal{H} \pm \mathcal{P}$ can easily be read off by comparing (2.19) and (2.24). At this point, it is useful to fix the freedom in the definition of $u, v$, and we will make the natural choice $\partial_u U = \partial_v V = 1$. With this choice, the field-dependent coordinates satisfy the following simple relation

$$u' \equiv \partial_\sigma u = 1 + \mu(\mathcal{H}-\mathcal{P}), \qquad v' \equiv \partial_\sigma v = 1 + \mu(\mathcal{H}+\mathcal{P}), \qquad (2.25)$$

where we used eq. (2.22). The radii of the field-dependent coordinates $u, v$ take the simple form

$$R_u \equiv \int_0^R d\sigma\, u' = R + \mu(H-P), \qquad R_v \equiv \int_0^R d\sigma\, v' = R + \mu(H+P), \qquad (2.26)$$

where $H = \oint d\sigma \mathcal{H}(\sigma)$ and $P = \oint d\sigma \mathcal{P}(\sigma)$ are the total Hamiltonian and momentum.

To compute the Poisson bracket algebra of the charges (2.24), we will need the algebra of the currents $\mathcal{H} \pm \mathcal{P}$ and those of the normalized state - dependent coordinates $\hat{u}, \hat{v}$. The Poisson brackets of the currents are determined by their Poisson brackets in the undeformed theory through (2.6)

$$\{\mathcal{H}^{(0)}, \tilde{\mathcal{H}}^{(0)}\} = \{\mathcal{P}, \tilde{\mathcal{P}}\} = (\mathcal{P}+\tilde{\mathcal{P}})\partial_\sigma \delta(\sigma-\tilde{\sigma}),$$

$$\{\mathcal{H}^{(0)}, \tilde{\mathcal{P}}\} = \{\mathcal{P}, \tilde{\mathcal{H}}^{(0)}\} = (\mathcal{H}^{(0)} + \tilde{\mathcal{H}}^{(0)})\delta_\sigma \delta(\sigma-\tilde{\sigma}), \qquad (2.27)$$

together with the exact expression (2.6) for the deformed Hamiltonian density. Here, we introduced the short-hand notation $\mathcal{H}(\sigma) \equiv \mathcal{H}, \mathcal{H}(\tilde{\sigma}) \equiv \tilde{\mathcal{H}}$ and similarly for $\mathcal{P}$. Explicitly, we find

$$\{\mathcal{H}, \tilde{\mathcal{H}}\} = \{\mathcal{P}, \tilde{\mathcal{P}}\} = (\mathcal{P}+\tilde{\mathcal{P}})\partial_\sigma \delta(\sigma-\tilde{\sigma}), \qquad \{\mathcal{H}, \tilde{\mathcal{P}}\} = \frac{\mathcal{H}^{(0)} + \tilde{\mathcal{H}}^{(0)} + 2\mu\mathcal{P}(\mathcal{P}+\tilde{\mathcal{P}})}{1+2\mu\mathcal{H}}\partial_\sigma \delta(\sigma-\tilde{\sigma}),$$
$$(2.28)$$

where $\mathcal{H}^{(0)} = \mathcal{H} + \mu(\mathcal{H}^2 - \mathcal{P}^2)$. The Poisson brackets of the field-dependent coordinates can be determined from the above though the identity (2.25), which shows that up to a factor of $\mu^2$, the Poisson brackets of the spatial derivatives $u', v'$ are identical to those of $\mathcal{H} \pm \mathcal{P}$. Explicitly,

$$\{u', \tilde{u}'\} = -2\mu^2(G_- + \tilde{G}_-)\partial_\sigma \delta(\sigma-\tilde{\sigma}), \qquad \{v', \tilde{v}'\} = 2\mu^2(G_+ + \tilde{G}_+)\partial_\sigma \delta(\sigma-\tilde{\sigma}),$$

$$\{u', \tilde{v}'\} = -\{v', \tilde{u}'\} = 2\mu^3 F'(\sigma)\delta(\sigma-\tilde{\sigma}), \qquad (2.29)$$

where the functions $F$ and $G_\pm$ are defined as

$$F \equiv \frac{\mathcal{H}^2 - \mathcal{P}^2}{1+2\mu\mathcal{H}}, \qquad G_\pm \equiv (\mathcal{H} \pm \mathcal{P})\frac{1+\mu(\mathcal{H} \pm \mathcal{P})}{1+2\mu\mathcal{H}}. \qquad (2.30)$$

To obtain the Poisson brackets of the coordinates $u, v$ themselves, we integrate eq. (2.29) with respect to $\sigma, \tilde{\sigma}$. We find

$$\begin{aligned}
\{u, \tilde{u}\} &= -2\mu^2[G_-(\sigma)\Theta(\tilde{\sigma}-\sigma) - G_-(\tilde{\sigma})\Theta(\sigma-\tilde{\sigma}) + D_-(\sigma) - D_-(\tilde{\sigma})], \\
\{v, \tilde{v}\} &= 2\mu^2[G_+(\sigma)\Theta(\tilde{\sigma}-\sigma) - G_+(\tilde{\sigma})\Theta(\sigma-\tilde{\sigma}) + D_+(\sigma) - D_+(\tilde{\sigma})], \\
\{u, \tilde{v}\} &= 2\mu^3[F(\sigma)\Theta(\tilde{\sigma}-\sigma) + F(\tilde{\sigma})\Theta(\sigma-\tilde{\sigma}) + A(\sigma) + B(\tilde{\sigma})],
\end{aligned} \qquad (2.31)$$

where $A, B, D_\pm$ are integration functions that we will constrain below, and the range of the $\sigma, \tilde{\sigma}$ coordinates is restricted to $[0, R)$.

When working with the above Poisson brackets, one should take into account the fact that the field-dependent coordinates wind around the spatial circle. There are two possible ways to deal with this issue: we either restrict the variables $\sigma, \tilde{\sigma}$ as above and carefully keep track of the winding of $u, v$ or, more elegantly, we consider them living on $\mathbb{R}$ instead of $[0, R)$ and make the Poisson brackets periodic by replacing $\delta \to \delta_R$, where $\delta_R(x) \equiv \sum_{n\in\mathbb{Z}} \delta(x+nR)$ is the "comb function" with periodicity $R$, and $\Theta \to \Theta_R(x) \equiv \int^x dy\, \delta_R(y)$. Note that $\Theta_R$ coincides with the usual Heaviside function as long as $|\sigma - \tilde{\sigma}| < R$, but counts the additional winding whenever the difference grows larger.

The Poisson brackets between the coordinates and the field-dependent radii (2.26) are determined from (2.31) by using the fact that $R_u = u(R) - u(0)$ for an arbitrary origin $\sigma = 0$. We obtain

$$\{u(\sigma), R_u\} = -2\mu^2(G_-(\sigma) + G_-(0) - w_{D_-}), \quad \{u(\sigma), R_v\} = 2\mu^3(F(\sigma) - F(0) + w_B),$$
$$\{v(\sigma), R_u\} = -2\mu^3(F(\sigma) - F(0) + w_A), \quad\quad \{v(\sigma), R_v\} = 2\mu^2(G_+(\sigma) + G_+(0) - w_{D_+}),$$
$$(2.32)$$

where the winding of a function $A(\sigma)$ is defined as

$$w_A \equiv A(R) - A(0). \tag{2.33}$$

Note the winding can in principle depend on the starting point, '0'. Since these Poisson brackets are periodic in $\sigma$, it is clear that the Poisson brackets of the radii with each other vanish, consistently with their definition (2.26) in terms of the total energy and momentum.

The functions $A, B, D_\pm$ are subject to several constraints, such as charge conservation. As detailed in appendix A.1, when the theory is on a compact space, this requires the windings to satisfy

$$w_{D_-} + \mu w_B = G_-(0) + \mu F(0),$$
$$w_{D_+} + \mu w_A = G_+(0) + \mu F(0). \tag{2.34}$$

Note in particular that the windings cannot be zero in finite size.

We obtain further constraints on the functions $A, B, D_\pm$ by requiring that the Poisson brackets (2.31) be consistent with time evolution. Parametrising $A, B, D_\pm$ in terms of four new functions $X, Y, \bar{X}, \bar{Y}$

$$A = F\left(Y - \frac{1}{2}\right) - G_-\bar{Y}, \quad\quad D_+ = G_+\left(Y - \frac{1}{2}\right) - \mu^2 F\bar{Y}, \tag{2.35}$$

$$B = F\left(\bar{X} - \frac{1}{2}\right) - G_+ X, \quad\quad D_- = G_-\left(\bar{X} - \frac{1}{2}\right) - \mu^2 F X, \tag{2.36}$$

time evolution, as we show in appendix A.2, restricts the above functions to satisfy

$$\dot{X} - \{H, X\} = \frac{1 + 2\mu\mathcal{P}}{1 + 2\mu\mathcal{H}} X', \quad\quad \dot{\bar{X}} - \{H, \bar{X}\} = -\frac{1 - 2\mu\mathcal{P}}{1 + 2\mu\mathcal{H}} \bar{X}' \tag{2.37}$$

and similarly for $Y, \bar{Y}$. This translates into the requirement that $X, Y$ be only functions of $u$, while $\bar{X}, \bar{Y}$ can only depend on $v$.

Additional constraints on the functions $A, B, D_\pm/X, \bar{X}, Y, \bar{Y}$ follow from the requirement that the Poisson brackets of the conserved charges yield other conserved charges. More precisely, as detailed in appendix A.3, the non-zero Poisson brackets of the radii $R_{u,v}$ with the energy currents induce terms proportional to $Q_{uf'}$ and $\bar{Q}_{v\bar{f}'}$. These terms are not conserved due to

the lack of periodicity of the associated functions and should thus be cancelled. This entirely fixes the winding part of the functions $X, Y, \bar{X}, \bar{Y}$, i.e.

$$X(\hat{u}) = X_p(\hat{u}), \quad \bar{X}(\hat{v}) = \hat{v} + \bar{X}_p(\hat{v}), \quad Y(\hat{u}) = \hat{u} + Y_p(\hat{u}), \quad \bar{Y}(\hat{v}) = Y_p(\hat{v}), \tag{2.38}$$

where the functions $X_p(\hat{u})$, etc., are periodic in their argument. The Poisson bracket algebra of the conserved charges is then given by

$$\begin{aligned}
\{Q_f, \bar{Q}_{\bar{f}}\} &= -\mu(Q_{\partial_u f} Q_{\bar{X}_p \partial_v \bar{f}} + Q_{Y_p \partial_u f} \bar{Q}_{\partial_v \bar{f}}), \\
\{Q_f, Q_g\} &= Q_{f \partial_u g - \partial_u f g} + \mu^2(Q_{\partial_u f} Q_{X_p \partial_u g} - Q_{X_p \partial_u f} Q_{\partial_u g}), \\
\{\bar{Q}_{\bar{f}}, \bar{Q}_{\bar{g}}\} &= \bar{Q}_{\partial_v \bar{f} \bar{g} - \bar{f} \partial_v \bar{g}} + \mu^2(\bar{Q}_{\partial_v \bar{f}} \bar{Q}_{\bar{Y}_p \partial_v \bar{g}} - \bar{Q}_{\bar{Y}_p \partial_v \bar{f}} \bar{Q}_{\partial_v \bar{g}}).
\end{aligned} \tag{2.39}$$

The details of this calculation are worked out in appendix A.3. Even more constraints on the functions $X_p, \bar{X}_p, Y_p, \bar{Y}_p$ could be derived from the Jacobi identities, but we will not pursue this avenue here.

Of course, in general we would expect the Poisson bracket of the field-dependent coordinates to be fixed by the natural interpretation of the integral of the left- and right-moving energy densities, $\mathcal{H} \pm \mathcal{P}$. As we will see in sections 3 and 4, this is indeed what happens in the case of $J\bar{T}$ and $JT_a$ - deformed CFTs, where the current is naturally represented in terms of the derivative of an auxiliary scalar $\varphi$, whose known Poisson brackets naturally fix the potential ambiguities coming from integrating the Poisson brackets of $\varphi'$. In the $T\bar{T}$ case, the existence of an analogous interpretation would determine the functions $X, \bar{X}, Y, \bar{Y}$. In the absence of such an interpretation, an obvious choice is to set the periodic part $X_p$, etc., of these arbitrary functions to zero, case in which the Poisson brackets take the very simple form

$$\begin{aligned}
\{u, \tilde{u}\} &= -2\mu^2\left[(G_- + \tilde{G}_-)\varepsilon(\tilde{\sigma} - \sigma) + \frac{G_- v - \tilde{G}_- \tilde{v}}{R_v}\right], \\
\{u, \tilde{v}\} &= 2\mu^3\left[(F - \tilde{F})\varepsilon(\tilde{\sigma} - \sigma) + \frac{Fu}{R_u} + \frac{\tilde{F}\tilde{v}}{R_v}\right], \\
\{v, \tilde{v}\} &= 2\mu^2\left[(G_+ + \tilde{G}_+)\varepsilon(\tilde{\sigma} - \sigma) + \frac{G_+ u - \tilde{G}_+ \tilde{u}}{R_u}\right],
\end{aligned} \tag{2.40}$$

where $\epsilon(\tilde{\sigma} - \sigma) = \Theta(\tilde{\sigma} - \sigma) - \frac{1}{2}$. Apart from the winding terms, these look rather naturally as the commutation relations of bosons. In the non-compact case, where the last terms drop out, we may be able to justify this as the correct, or the most natural, choice. In the compact case, however, consistency of the charge algebra requires the winding terms to be present in the Poisson brackets of the field-dependent coordinates, and so far we do not have a good interpretation for them. In any case, the choice (2.40) implies that the Poisson brackets of the charges organise into two commuting copies of a functional Witt algebra,

$$\{Q_f, \bar{Q}_{\bar{f}}\} = 0, \qquad \{Q_f, Q_g\} = Q_{f \partial_u g - \partial_u f g}, \qquad \{\bar{Q}_{\bar{f}}, \bar{Q}_{\bar{g}}\} = \bar{Q}_{\partial_v \bar{f} \bar{g} - \bar{f} \partial_v \bar{g}}, \tag{2.41}$$

i.e., a Witt algebra at the level of the functions that label the conserved charges.

On the cylinder, however, it is customary to write the charge algebra in terms of the Fourier modes, $L_n, \bar{L}_n$ obtained by choosing the basis of functions $f_n(\hat{u}) = e^{in\hat{u}}$ and $\bar{f}_n(\hat{v}) = e^{-in\hat{v}}$ and multiplying all charges by a factor of $R$, so as to make them dimensionless. Then, the classical charge algebra (2.41) translates into

$$\{L_m, L_n\} = -i\frac{m-n}{1 + 2\hat{\mu}\bar{L}_0}L_{m+n}, \quad \{\bar{L}_m, \bar{L}_n\} = -i\frac{m-n}{1 + 2\hat{\mu}L_0}\bar{L}_{m+n}, \quad \{L_m, \bar{L}_n\} = 0, \tag{2.42}$$

where $\hat{\mu} = \mu/R^2$ is the dimensionless $T\bar{T}$ coupling on the cylinder and the denominators come from the ratios $R/R_{u,v}$ that now appear explicitly in the charge algebra. Thus, while we do obtain a Witt algebra at the level of the functions that parametrize the conserved charges, it is not the usual Witt algebra when written in terms of Fourier modes.[5]

So far, we have been concentrating exclusively on the compact case. The discussion and calculations we performed in this section and in appendix A all simplify if the theory lives on the plane $R \to \infty$, assuming that the energy and momentum densities decay at spatial infinity. First, the functions $f$ and $\bar{f}$ in eq. (2.24) become just functions of $u$ and $v$, which are assumed to decay or stay constant at infinity, so that the associated charges are finite. The fall-off conditions on $\mathcal{H}, \mathcal{P}$ automatically ensure charge conservation, so we can simply drop all the winding terms that complicated the compact case. In particular, the unintuitive winding terms proportional to $u/R_u$ and $v/R_v$ in the commutator (2.40) can be dropped, as well as those in (2.38) and (A.26)–(A.28), who were related to charge non-conservation. Finally, in the non-compact setting only the functional form of the Witt algebra is relevant, which in this case is identical to its CFT counterpart.

## 2.3. $T\bar{T}$ - deformed free boson example

At the start of this section, we used the Lagrangian formulation to exhibit the infinite pseudo-conformal symmetries of $T\bar{T}$ - deformed CFTs. The ratios $\mathcal{L}, \bar{\mathcal{L}}$ of the stress tensor components played a special role in our analysis, acting as the (anti-)holomorphic stress tensor components of an auxiliary 'usual CFT' living in the $u, v$ coordinates. Even though the charges assumed a suggestive CFT-like form, (2.23), in terms of $\mathcal{L}, \bar{\mathcal{L}}$, the subsequent Poisson bracket analysis made no use of this structure.

In this section, we would like to discuss the relation between the $T\bar{T}$ - deformed and the original CFT in the Hamiltonian formalism. We concentrate on the special case of the free boson, where everything can be made entirely explicit. In the Hamiltonian formalism, the $T\bar{T}$-deformed and original CFT are related by a canonical transformation, as was nicely explained in [13] which can be spelled out explicitly for the free boson. As we will show, the reason that the $\mathcal{L}, \bar{\mathcal{L}}$ variables are not particularly useful for computing the equal-time Poisson brackets in the deformed theory is that this canonical transformation does not map equal times to each other. One would thus need to use Hamiltonian evolution inside the bracket in order to relate their Poisson brackets to those of the simple CFT obtained from (2.27).

The goal of this section is thus to show that the equal-time Poisson brackets of the quantities $\mathcal{L}, \bar{\mathcal{L}}$ in the $T\bar{T}$ - deformed CFT simply correspond to the unequal-time Poisson brackets of $\mathcal{H}^{(0)} \pm \mathcal{P}$ in the undeformed CFT, obtained by evolving with the equations of motion. We also use this opportunity to give a few explicit formulae for the free boson in our conventions and underline certain additional points, in both the Lagrangian and the Hamiltonian formalisms.

**Lagrangian analysis**

The action of a $T\bar{T}$ - deformed free boson is

$$S_L(\mu) = \frac{1}{2\mu} \int dt d\sigma \left( 1 - \sqrt{1 + 2\mu(\phi'^2 - \dot{\phi}^2)} \right). \tag{2.43}$$

---

[5] One can obtain the usual Witt algebra at the level of the Fourier modes by multiplying the left-moving charges by $R_u$ and the right-moving ones by $R_v$, as was done in e.g. [28]. One disadvantage of doing so is that the new $L_0$ and $\bar{L}_0$ would no longer represent the energy plus/minus the momentum of the state. Also, since the field-dependent radii have non-zero Poisson brackets with the charges of the opposite chirality, a non-zero commutator between the left and right-moving charges would be induced.

The components of the stress tensor are

$$T_{UU} = \frac{(\partial_U \phi)^2}{\sqrt{1 + 8\mu\, \partial_U \phi\, \partial_V \phi}} \,,$$

$$T_{VV} = \frac{(\partial_V \phi)^2}{\sqrt{1 + 8\mu\, \partial_U \phi\, \partial_V \phi}} \,,$$

$$T_{VU} = \frac{1}{4\mu} \left( 1 - \frac{1 + 4\mu\partial_U \phi\, \partial_V \phi}{\sqrt{1 + 8\mu\, \partial_U \phi\, \partial_V \phi}} \right). \tag{2.44}$$

It is easy to check that the trace identity (2.2) holds off-shell. Using the definition (2.9) of $\mathcal{L}, \bar{\mathcal{L}}$ and the above explicit form of the stress tensor, one can solve for the derivatives of $\phi$ in terms of $\mathcal{L}, \bar{\mathcal{L}}$

$$\partial_U \phi = \sqrt{\frac{\mathcal{L}}{2}} \frac{1}{1 - \mu\sqrt{\mathcal{L}\bar{\mathcal{L}}}}\,, \qquad \partial_V \phi = \sqrt{\frac{\bar{\mathcal{L}}}{2}} \frac{1}{1 - \mu\sqrt{\mathcal{L}\bar{\mathcal{L}}}}. \tag{2.45}$$

Using this, one can easily express the partial derivatives of the field $\phi$ with respect to $u, v$ in terms of $\mathcal{L}, \bar{\mathcal{L}}$

$$\partial_u \phi = \partial_u U(\partial_U \phi - \mu\mathcal{L}\partial_V \phi) = \partial_u U\sqrt{\frac{\mathcal{L}}{2}}\,, \qquad \partial_v \phi = \partial_v V(\partial_V \phi - \mu\bar{\mathcal{L}}\partial_U \phi) = \partial_v V\sqrt{\frac{\bar{\mathcal{L}}}{2}}, \tag{2.46}$$

which relates them (off-shell) to the stress tensor components of a free boson in the $u, v$ coordinates. The equation of motion for the $T\bar{T}$-deformed free boson reduces to the free wave equation $\partial_u \partial_v \phi = 0$ in terms of the field-dependent coordinates. Notice however that the coordinate transformation we have chosen *does not* map the action (2.43) to the free boson action in $u, v$ coordinates.

It is interesting to note that in the simple case of the $T\bar{T}$ - deformed free boson, an explicit off-shell solution for $\partial_u U$ and $\partial_v V$ can be found. A reason for the simplification is the off-shell compatibility relation $\partial_U \partial_V \phi = \partial_V \partial_U \phi$, which translates, using (2.45) and $\partial_U - \mu\mathcal{L}\partial_V = (\partial_u U)^{-1}\partial_u$ , $\partial_V - \mu\bar{\mathcal{L}}\partial_U = (\partial_v V)^{-1}\partial_v$, into a constraint on $\mathcal{L}, \bar{\mathcal{L}}$, namely $\partial_u U\partial_v\sqrt{\mathcal{L}} = \partial_v V\partial_u\sqrt{\bar{\mathcal{L}}}$. Combining this with (2.16), we obtain an off-shell equation for $U(u, v)$

$$\partial_v(\partial_u U) = -2\mu\partial_u U \frac{\sqrt{\bar{\mathcal{L}}}\partial_v \mathcal{L}}{1 + \mu\sqrt{\mathcal{L}\bar{\mathcal{L}}}} \quad \Rightarrow \quad \partial_u U = U_0'(u)\exp\left(-2\mu\int_0^v dv\, \frac{\sqrt{\bar{\mathcal{L}}}\partial_v \mathcal{L}}{1 + \mu\sqrt{\mathcal{L}\bar{\mathcal{L}}}}\right). \tag{2.47}$$

This expression makes it clear that, while we do have a function worth of freedom in choosing the initial condition $U_0(u)$, no particular choice of this function will reduce to a simple choice for $\partial_u U$ off-shell. Similar comments hold for $\partial_v V$.

In the case of the free boson, there is a $U(1)$ current associated with the shift symmetry of $\phi$. Its components are

$$J_U = \frac{\partial_U \phi}{\sqrt{1 + 8\mu\partial_U \phi\, \partial_V \phi}}\,, \qquad J_V = \frac{\partial_V \phi}{\sqrt{1 + 8\mu\partial_U \phi\, \partial_V \phi}} \,. \tag{2.48}$$

There is also a current $\tilde{J}_U = \partial_U \phi$, $\tilde{J}_V = -\partial_V \phi$, which is topologically conserved. The combinations $J^\pm = (J + \tilde{J})/2$ are chiral and respectively anti-chiral in the undeformed CFT, leading to an infinite set of conserved charges. After the deformation, these currents are no longer chiral, but it can be easily checked (using (2.45)) that their components satisfy the relation

$$\frac{J_V^+}{J_U^+} = -\mu\bar{\mathcal{L}}\,, \qquad \frac{J_U^-}{J_V^-} = -\mu\mathcal{L}, \tag{2.49}$$

which is the same relation as that satisfied by the stress tensor components (2.10). Following our argument in the introduction, this implies that for any function $\chi(u)$ and $\bar{\chi}(v)$ which depend on the *same* field-dependent coordinates $u, v$ as the pseudo-conformal symmetries, the currents $\chi(u)J^+$ and $\bar{\chi}(v)J^-$ will be conserved. They generate an infinite set of affine $U(1)$ symmetries. Thus, the full symmetry of the original free CFT is still present after the deformation.

**Hamiltonian analysis**

The canonical momentum conjugate to $\phi$ in the $T\bar{T}$-deformed theory is

$$\pi = \frac{\dot{\phi}}{\sqrt{1 + 2\mu\phi'^2 - 2\mu\dot{\phi}^2}} \tag{2.50}$$

and the Hamiltonian density reads

$$\mathcal{H} = \pi\dot{\phi} - \mathcal{L} = \frac{1}{2\mu}\sqrt{1 + 2\mu(\pi^2 + \phi'^2) + 4\mu^2(\pi\phi')^2} - \frac{1}{2\mu}, \qquad \mathcal{P} = \pi\phi'. \tag{2.51}$$

Notice this is precisely of the form (2.6), derived in [13]. The equal-time Poisson bracket is

$$\{\phi(\sigma), \pi(\tilde{\sigma})\} = \delta(\sigma - \tilde{\sigma}). \tag{2.52}$$

It is useful to rewrite the canonical variables in terms of $\mathcal{L}, \bar{\mathcal{L}}$. We find

$$\pi = \frac{\partial_U\phi - \partial_V\phi}{\sqrt{1 + 8\mu\partial_U\phi\partial_V\phi}} = \frac{\sqrt{\mathcal{L}} - \sqrt{\bar{\mathcal{L}}}}{\sqrt{2}(1 + \mu\sqrt{\mathcal{L}\bar{\mathcal{L}}})}, \qquad \phi' = \partial_U\phi + \partial_V\phi = \frac{\sqrt{\mathcal{L}} + \sqrt{\bar{\mathcal{L}}}}{\sqrt{2}(1 - \mu\sqrt{\mathcal{L}\bar{\mathcal{L}}})}. \tag{2.53}$$

We subsequently notice that the combinations

$$\frac{\pi + \phi'}{2} = \sqrt{\frac{\mathcal{L}}{2}}\frac{1 + \mu\bar{\mathcal{L}}}{1 - \mu^2\mathcal{L}\bar{\mathcal{L}}} = \partial_\sigma u\,\partial_u\phi, \qquad \frac{\phi' - \pi}{2} = \sqrt{\frac{\bar{\mathcal{L}}}{2}}\frac{1 + \mu\mathcal{L}}{1 - \mu^2\mathcal{L}\bar{\mathcal{L}}} = \partial_\sigma v\,\partial_v\phi, \tag{2.54}$$

where we used (2.22) and (2.46) in the last step. These relations hold off-shell and independently of what we choose $\partial_u U$ to be. The quantities $\partial_u\phi$ and $\partial_v\phi$ simply represent the holomorphic and anti-holomorphic derivative of a free boson living in the auxiliary $u, v$ coordinate system. Letting $u, v = \sigma_c \pm t_c$, where the subscript $c$ stands for "auxiliary CFT", these derivatives can be further written in terms of the canonical momentum of the auxiliary free boson as

$$\partial_u\phi = \frac{\pi_c + \phi'_c}{2}, \qquad \partial_v\phi = \frac{\phi'_c - \pi_c}{2}, \tag{2.55}$$

where $\pi_c = \partial_{t_c}\phi_c$ is the canonical momentum. Then, (2.54) leads to the following map between the free boson canonical variables and the $T\bar{T}$ - deformed ones

$$\phi_c(u(\sigma), v(\sigma)) = \phi(\sigma), \qquad \pi_c(u(\sigma), v(\sigma)) = \frac{\pi + \phi'}{2u'} + \frac{\pi - \phi'}{2v'}, \tag{2.56}$$

which holds on a constant $t$ slice in the $T\bar{T}$ - deformed CFT. One can easily check that $\phi' = u'\partial_u\phi_c + v'\partial_v\phi_c$ agrees with the expectation from (2.54) via (2.55). Upon choosing

the gauge[6] $\partial_u U = \partial_v V = 1$, the spatial derivatives $u', v'$ are given by (2.25). This map is very similar to that presented in [13], who made a slightly different choice of gauge. Since both the free boson and the $T\bar{T}$ - deformed free boson can be obtained from different gauge-fixings of the three-dimensional Nambu–Goto action, the map (2.56) between the two is canonical.

Since the map is canonical, it should preserve the Poisson brackets. Let us compute, for concreteness, the Poisson bracket of $\pi_c$ with itself, using the right-hand side of (2.56). The result can be written in the suggestive form[7]

$$\{\pi_c(\sigma), \pi_c(\tilde{\sigma})\} = \frac{1}{2}\left(\frac{1}{u'\tilde{u}'} - \frac{1}{v'\tilde{v}'}\right)\partial_\sigma \delta(\sigma - \tilde{\sigma}). \tag{2.58}$$

The fact that the equal-time Poisson bracket of the two free boson momenta does not vanish can be understood intuitively from the fact that an equal time slice in the $T\bar{T}$ - deformed system does not generally map to an equal time slice in the free boson one. We should thus compare this with the *unequal* time Poisson bracket in the free boson system, obtained from the relations

$$\{\pi_+^c(u), \pi_+^c(\tilde{u})\} = \frac{1}{2}\partial_u \delta(u - \tilde{u}), \quad \{\pi_-^c(v), \pi_-^c(\tilde{v})\} = -\frac{1}{2}\partial_v \delta(v - \tilde{v}), \quad \{\pi_+^c(u), \pi_-^c(\tilde{v})\} = 0, \tag{2.59}$$

where $\pi_\pm^c = (\pi_c \pm \phi_c')/2$. These commutators follow from the equations of motion, which set $\pi_+^c$ to be only a function of $u$, and $\pi_-^c$ only a function of $v$. Subtracting the first two terms to obtain the $\pi_c$ Poisson bracket and changing variables from $u, v$ to $\sigma$ yields precisely (2.58). A similar argument holds for the Poisson bracket of $\phi_c'$ with itself.

More interesting is the result for $\{\pi_+^c, \pi_-^c\}$, which in the free boson system vanishes at all times; however the explicit calculation using (2.56) shows that in our case, it does not. Instead, it takes the form

$$\{\pi_+^c(\sigma), \pi_-^c(\tilde{\sigma})\} = \frac{1}{u'v'}\partial_\sigma \ln\frac{1 + 2\mu\phi'^2}{1 + 2\mu\pi^2}\delta(\sigma - \tilde{\sigma}), \tag{2.60}$$

where we used (C.6) to rewrite the distribution in a more convenient form. This is nevertheless still consistent with the Poisson brackets of the free boson system, since we should compare the Poisson brackets of the full momenta, $\Pi_+^c = \int du\, \pi_+^c(u)$ and $\Pi_-^c = \int dv\, \pi_-^c(v)$, rather than those of the momentum densities. The right-hand side of (2.60) is then just a total derivative, which integrates to zero.

It is easy to show that the Poisson brackets of $\mathcal{L}, \bar{\mathcal{L}}$, which we had identified with the components of the stress tensor of an auxiliary CFT, similarly correspond to CFT commutators at unequal times. This holds not only for the free boson, but more generally in any $T\bar{T}$ - deformed CFT. To show this, we first express $\mathcal{L}, \bar{\mathcal{L}}$ in terms of $\mathcal{H}^{(0)}, \mathcal{P}$ via

$$\mathcal{L} = \frac{\mathcal{H} + \mathcal{P}}{1 + \mu(\mathcal{H} - \mathcal{P})}, \quad \bar{\mathcal{L}} = \frac{\mathcal{H} - \mathcal{P}}{1 + \mu(\mathcal{H} + \mathcal{P})}, \tag{2.61}$$

by using the explicit form (2.6) for $\mathcal{H}$. Then, we use the undeformed Poisson brackets (2.27) to compute the Poisson brackets of $\mathcal{L}, \bar{\mathcal{L}}, \tilde{\mathcal{L}}, \tilde{\bar{\mathcal{L}}}$, obtaining

$$\{\mathcal{L}, \tilde{\mathcal{L}}\} = \frac{2(\mathcal{L} + \tilde{\mathcal{L}})}{u'\tilde{u}'}\partial_\sigma \delta(\sigma - \tilde{\sigma}), \qquad \{\bar{\mathcal{L}}, \tilde{\bar{\mathcal{L}}}\} = -\frac{2(\bar{\mathcal{L}} + \tilde{\bar{\mathcal{L}}})}{v'\tilde{v}'}\partial_\sigma \delta(\sigma - \tilde{\sigma}), \tag{2.62}$$

---

[6]The terminology "gauge choice" refers to the fact that the $T\bar{T}$ - deformed free boson corresponds to the Nambu–Goto action in static gauge, whereas the usual free boson is recovered from the Nambu–Goto dynamics in conformal gauge. Since the conformal gauge condition is not sufficient to fix the reparametrization symmetry of the Nambu–Goto action, one needs additional conditions, such as the one we use.

[7]For this calculation, it is useful to note that for two functions only depending on $\pi$ and $\phi'$, the Poisson bracket is

$$\{f(\sigma), g(\tilde{\sigma})\} = [\partial_{\phi'}f(\sigma)\partial_\pi g(\tilde{\sigma}) + \partial_\pi f(\sigma)\partial_{\phi'}g(\tilde{\sigma})]\partial_\sigma \delta(\sigma - \tilde{\sigma}). \tag{2.57}$$

The result is further simplified using the equivalence of distributions (C.3).

$$\{\mathcal{L}, \tilde{\bar{\mathcal{L}}}\} = \frac{4}{\mu u' v'} \partial_\sigma \ln(1 - \mu^2 \mathcal{L}\bar{\mathcal{L}}) \delta(\sigma - \tilde{\sigma}). \tag{2.63}$$

The first two Poisson brackets precisely correspond to an unequal-time Poisson bracket in a CFT, thus confirming our intuition. The last Poisson bracket does not obviously match an appropriate unequal-time CFT commutator, but its form suggests it can be dealt with in a similar manner as the result for $\{\pi_+^c, \pi_-^c\}$ in (2.60). However, (2.62) is not sufficient, as one may have hoped, to give a one-line derivation of the charge algebra if the charges are written in the form (2.23). The reason is that, as we saw in the previous subsection, the $\mathcal{L}, \bar{\mathcal{L}}$ brackets are but one of several contributions to the conserved charges. While the charge algebra (2.41) is expected to emerge also from this slightly different way of performing the calculation, the use of the $\mathcal{L}, \bar{\mathcal{L}}$ variables does not appear to bring any particular simplification to the computation of the Poisson brackets of the charges, as compared to the calculation performed in the previous subsection.

# 3. $J\bar{T}$-deformed CFTs

The $J\bar{T}$ deformation [55] is an irrelevant deformation of two-dimensional QFTs of the Smirnov–Zamolodchikov type [2], constructed from the components of a $U(1)$ current and those of the generator, $T_{\alpha V}$, of right-moving translations. In Lorentzian signature, the flow of the action is given by

$$\frac{\partial S_{J\bar{T}}}{\partial \lambda} = \int d^2 x \sqrt{\gamma} \epsilon^{\alpha\beta} J_\alpha T_{\beta V} = -\int dU dV (J_U T_{VV} - J_V T_{UV}), \tag{3.1}$$

where, as before, $U, V = \sigma \pm t$ are null coordinates on two-dimensional Minkowski space. The current $J$ does not need to be chiral along the flow.

When the seed theory is a CFT, the finite-size spectrum of the deformed theory can be computed exactly. Introducing the right-moving energy $E_R = (E - P)/2$, its eigenvalues in the deformed CFT are given by [59,69]

$$E_R(\lambda, R) = \frac{4\pi}{\lambda^2 k} \left( R - \lambda Q_+^{(0)} - \sqrt{\left(R - \lambda Q_+^{(0)}\right)^2 - \frac{\lambda^2 k}{2\pi} E_R^{(0)} R} \right), \tag{3.2}$$

where $E_R^{(0)}$ is right-moving energy of the corresponding state in the undeformed CFT, $R$ is the circumference of the circle, and $Q_+^{(0)}$ is the charge associated with the chiral part, $J^+$, of the current. The chiral charge flows as

$$Q_+ = Q_+^{(0)} + \frac{\lambda k}{4\pi} E_R. \tag{3.3}$$

It can easily be checked that the combination $E_L - 2\pi Q_+^2/(kR)$ is constant along the flow, where $E_L$ is the left-moving energy $E_L = (E + P)/2$. This suggests an interpretation of the $J\bar{T}$ deformation as spectral flow in the left-moving sector, where the spectral flow parameter is proportional to the right-moving energy of the state.

A nice feature of the $J\bar{T}$ deformation is that, when the seed theory is a CFT, it preserves an $SL(2, \mathbb{R})_L \times U(1)_R$ subgroup of the original conformal group. As a consequence, the deformed CFT remains local and conformal on the left, while becoming non-local on the right. This remaining conformal structure allows for an analysis of correlation functions [61]. It is to be expected that the left-moving conformal symmetries, together with the left-moving $U(1)$ symmetries, are enhanced to a full Virasoro $\ltimes$ Kac-Moody algebra. This is precisely what

was found in the holographic analysis of [70], which provided a holographic dual to the $J\bar{T}$ deformation in terms of AdS$_3$ gravity coupled to a $U(1)$ Chern–Simons gauge field with mixed boundary conditions at infinity and spelled out the asymptotic symmetries associated with these boundary conditions. In addition, the asymptotic symmetry analysis of [70] showed that the right-moving $U(1)$ translational symmetries of the system are also enhanced to a right-moving Virasoro symmetry, but one that is generated by field-dependent diffeomorphisms and gauge transformations. While the holographic setup of [70] assumed that $J$ was chiral, we will see in this section that for non-chiral $J$, also its right-moving part generates an infinite-dimensional, field-dependent Kac-Moody algebra.

In this section, we would like to exhibit the two-fold Virasoro $\ltimes$ Kac-Moody symmetries of $J\bar{T}$ - deformed CFTs from a canonical analysis, which will complement the holographic results of [70]. To warm up, we will start with an analysis of the $J\bar{T}$ - deformed free boson, first in the Lagrangian and then in the Hamiltonian formalism. We then develop a general Hamiltonian framework, along the lines of [13], for $J\bar{T}$ - deformed classical CFTs, use it to construct conserved charges and show that the presence of two commuting copies of the Witt - Kac-Moody algebra is a completely general feature of $J\bar{T}$ - deformed CFTs.

### 3.1. $J\bar{T}$-deformed free boson: infinite symmetries

The $J\bar{T}$ - deformed free boson was first studied in [55]. The $SL(2,\mathbb{R})_L \times U(1)_R$ symmetry of the problem require the action to take the form

$$S_{J\bar{T}} = -\int dU dV \, \partial_U \phi \, \partial_V \phi \, \mathcal{F}(\lambda \partial_V \phi), \tag{3.4}$$

for some function $\mathcal{F}$. The $J\bar{T}$ flow equation (3.1) determines it to be

$$\mathcal{F}(x) = \frac{1}{1-x}. \tag{3.5}$$

The equations of motion are simply

$$\partial_V \left( \frac{\partial_U \phi}{1 - \lambda \partial_V \phi} \right) = 0, \tag{3.6}$$

with general solution $\phi(U,V) = f(U) + g(V - \lambda f(U))$, for two arbitrary functions $f, g$.

One can easily show that the action (3.4) is equivalent *off-shell* to a free boson, via the field-dependent coordinate transformation[8]

$$u = U, \qquad v = V - \lambda \phi(U,V). \tag{3.7}$$

Indeed, using the relation between the first derivatives of $\phi$

$$\partial_U \phi = \frac{\partial_u \phi}{1 + \lambda \partial_v \phi}, \qquad \partial_V \phi = \frac{\partial_v \phi}{1 + \lambda \partial_v \phi}, \tag{3.8}$$

one can easily check that the action (3.4) becomes $\int du dv \, \partial_u \phi \, \partial_v \phi$.

As in the case of the $T\bar{T}$ deformation, the field-dependent coordinates in terms of which the dynamics reduces to that of the undeformed CFT are also the ones appearing in the infinite-dimensional pseudo-conformal symmetries. To see this, let us compute the components of the

---

[8]In this section, as well as in the next one, we will abusively be using the same notation $u, v$ for the field-dependent coordinates, even though their expression in terms of the fields changes from (2.12) in $T\bar{T}$, to (3.7) in $J\bar{T}$, to (4.13) in $JT_a$ and (4.21) in $\tilde{J}T_a$. We hope that this choice, made for ease of notation, will not cause confusion to the reader.

stress tensor

$$T_{UU} = \frac{(\partial_U \phi)^2}{(1-\lambda\partial_V\phi)^2}, \quad T_{VU} = 0, \quad T_{UV} = \frac{\lambda\partial_U\phi(\partial_V\phi)^2}{(1-\lambda\partial_V\phi)^2}, \quad T_{VV} = \frac{(\partial_V\phi)^2}{1-\lambda\partial_V\phi}. \tag{3.9}$$

The stress tensor is not symmetric because the $J\bar{T}$ deformation breaks Lorentz invariance, and the $T_{VU}$ component vanishes as a consequence of the $SL(2,\mathbb{R})$ symmetry preserved by the deformation. Noting that

$$\frac{\partial_U v}{\partial_V v} = -\frac{\lambda\partial_U\phi}{1-\lambda\partial_V\phi} = -\frac{T_{UV}}{T_{VV}} \tag{3.10}$$

and using (1.2), we conclude that the action is invariant under

$$U \to U + \epsilon f(U), \quad V \to V - \epsilon\bar{f}(v) \tag{3.11}$$

for arbitrary functions $f, \bar{f}$ of their respective arguments. The first transformation is simply the infinite enhancement of the left-moving conformal symmetries. The second transformation corresponds to a field-dependent, right-moving pseudo-conformal transformation.

The conserved energy-momentum charges are given by (2.18), and read

$$Q_f = \int d\sigma f(U)T_{UU}, \quad \bar{Q}_{\bar{f}} = \int d\sigma\bar{f}(v)(T_{VV} - T_{UV}). \tag{3.12}$$

Thus, we find that the $J\bar{T}$ - deformed free boson possesses two infinite sets of conserved charges, one associated to a usual left-moving conformal symmetry, and the other one associated with a field-dependent right-moving coordinate transformation.

There are additional charges corresponding to the $U(1)$ currents. The $U(1)$ current associated to the shift symmetry of $\phi$ is

$$J_U = \frac{\partial_U\phi}{(1-\lambda\partial_V\phi)^2}, \quad J_V = \frac{\partial_V\phi}{1-\lambda\partial_V\phi}. \tag{3.13}$$

There is additionally a topologically conserved current, $\tilde{J} = \star d\phi$, with components

$$\tilde{J}_U = \partial_U\phi, \quad \tilde{J}_V = -\partial_V\phi. \tag{3.14}$$

In a CFT, the combinations $J^\pm = (J \pm \tilde{J})/2$ are chiral and respectively anti-chiral, and generate two copies of a $U(1)$ Kac-Moody algebra, one left-moving and one right-moving. Due to the $SL(2,\mathbb{R})_L$ symmetry of the deformed theory, we would still expect [71] to be able to construct a purely left-moving conserved current. Indeed, it is not hard to notice that the combination

$$K_\alpha \equiv \frac{1}{2}(J_\alpha + \tilde{J}_\alpha - \lambda T_{\alpha V}) \tag{3.15}$$

has a vanishing right-moving component. Concretely,

$$K_U = \frac{\partial_U\phi}{1-\lambda\partial_V\phi}, \quad K_V = 0. \tag{3.16}$$

The left-moving component is holomorphically conserved, as seen by using the equations of motion (3.6). We also note that for the single boson $T_{UU} = K_U^2$, so the combination $T_{UU} - K_U^2$ is trivially independent of $\lambda$. As we will see, the $\lambda$ - independence of this combination generalizes and corresponds to the fact that the $J\bar{T}$ deformation induces a kind of momentum-dependent spectral flow transformation on the left-movers [61,69].

Since the $V$ component of the current $K_\alpha$ vanishes, the current $\chi(U)K_\alpha$ is conserved for any function $\chi(U)$. Thus, we find an infinite set of conserved charges associated with this chiral $U(1)$ symmetry, given by

$$P_\chi = \int d\sigma \, \chi(U) K_U. \tag{3.17}$$

It is interesting to ask whether there also exists an infinite-dimensional enhancement of the right-moving $U(1)$ symmetry. A natural candidate is the current $J^-$, which is purely right-moving in the undeformed CFT. In presence of the deformation, its components are

$$J_U^- = \frac{\lambda \partial_U \phi \, \partial_V \phi}{(1 - \lambda \partial_V \phi)^2}\left(1 - \frac{\lambda}{2}\partial_V \phi\right), \quad J_V^- = \frac{\partial_V \phi}{1 - \lambda \partial_V \phi}\left(1 - \frac{\lambda}{2}\partial_V \phi\right). \tag{3.18}$$

These have the important property that

$$\frac{J_U^-}{J_V^-} = -\frac{\partial_U v}{\partial_V v}, \tag{3.19}$$

which implies that the current $\bar{\chi}(v)J_\alpha^-$ is conserved for any function $\bar{\chi}(v)$. In fact, any linear combination of $J_\alpha^-$ and $T_{\alpha V}$, when multiplied by an arbitrary function of $v$, will give rise to a conserved current. The conserved charges associated with the $\bar{\chi}(v)J_\alpha^-$ currents are

$$\bar{P}_{\bar{\chi}} = \int d\sigma \, \bar{\chi}(v)(J_U^- - J_V^-). \tag{3.20}$$

## 3.2. Hamiltonian analysis and charge algebra

Given the conserved charges (3.12), (3.17) and (3.20), we would now like to compute their Poisson bracket algebra. For this, we should first pass to the Hamiltonian formulation of the $J\bar{T}$ - deformed free boson.

The momentum conjugate to $\phi$ is

$$\pi \equiv \frac{\partial \mathcal{L}}{\partial \dot{\phi}} = \dot{\phi}\mathcal{F} - \frac{\lambda}{4}(\dot{\phi}^2 - \phi'^2)\mathcal{F}'. \tag{3.21}$$

Using the explicit form (3.5) of $\mathcal{F}$, one can find the expression for $\dot{\phi}$ in terms of $\pi$ and $\phi'$

$$\dot{\phi} = \phi' - \frac{2}{\lambda}\left(1 - \sqrt{\frac{1 - \lambda \phi'}{1 - \lambda \pi}}\right). \tag{3.22}$$

The Hamiltonian density reads

$$\mathcal{H} = \pi \dot{\phi} - \mathcal{L} = -\frac{4}{\lambda^2}\sqrt{(1 - \lambda \pi)(1 - \lambda \phi')} + \left(\phi' - \frac{2}{\lambda}\right)\left(\pi - \frac{2}{\lambda}\right). \tag{3.23}$$

For reasons that will soon become clear, it is convenient to introduce the left- and right-moving energy and current densities

$$\mathcal{H}_{L,R} = \frac{\mathcal{H} \pm \mathcal{P}}{2}, \qquad \mathcal{J}_\pm = \frac{\pi \pm \phi'}{2}, \tag{3.24}$$

where the momentum density $\mathcal{P} = \pi \phi'$, and write all quantities of interest in terms of the right-moving energy current, $\mathcal{H}_R$. The expression for $\mathcal{H}_R$ is then

$$\mathcal{H}_R = \frac{2}{\lambda^2}\left(1 - \lambda \mathcal{J}_+ - \sqrt{(1 - \lambda \mathcal{J}_+)^2 - \lambda^2 \mathcal{H}_R^{(0)}}\right), \tag{3.25}$$

where the undeformed right-moving Hamiltonian current $\mathcal{H}_R^{(0)}$ is

$$\mathcal{H}_R^{(0)} = \frac{\mathcal{H}^{(0)} - \mathcal{P}}{2} = \frac{(\pi - \phi')^2}{4}. \tag{3.26}$$

The expression for the conserved charges (3.12) in terms of $\mathcal{H}_R, \pi$ and $\phi'$ is

$$Q_f = \int d\sigma f(U)\mathcal{H}_L(\sigma), \qquad \bar{Q}_{\bar{f}} = \int d\sigma \bar{f}(v)\mathcal{H}_R(\sigma), \tag{3.27}$$

since, by definition, $\mathcal{H}_R = -T_{tV} = T_{VV} - T_{UV}$, and $\mathcal{H}_L = T_{tU} = T_{UU}$ is the generator of left-moving translations. The $U(1)$ charges (3.17) and (3.20) read

$$P_\chi = \int d\sigma \, \chi(U)\left(\mathcal{J}_+ + \frac{\lambda \mathcal{H}_R}{2}\right), \qquad \bar{P}_{\bar{\chi}} = \int d\sigma \, \bar{\chi}(v)\,\mathcal{J}_-. \tag{3.28}$$

It is useful to compute the Poisson brackets of the above symmetry generators in two steps. First, we use the free-field expression (3.26) for the undeformed Hamiltonian and the currents to show that

$$\{\mathcal{H}_R^{(0)}(\sigma), \mathcal{H}_R^{(0)}(\tilde{\sigma})\} = -\{\mathcal{H}_R^{(0)}(\sigma), \mathcal{P}(\tilde{\sigma})\} = -\left(\mathcal{H}_R^{(0)}(\sigma) + \mathcal{H}_R^{(0)}(\tilde{\sigma})\right)\partial_\sigma \delta(\sigma - \tilde{\sigma}),$$

$$\{\mathcal{P}(\sigma), \mathcal{P}(\tilde{\sigma})\} = (\mathcal{P}(\sigma) + \mathcal{P}(\tilde{\sigma}))\partial_\sigma \delta(\sigma - \tilde{\sigma}), \qquad \{\mathcal{J}_\pm(\sigma), \mathcal{J}_\pm(\tilde{\sigma})\} = \pm\frac{1}{2}\partial_\sigma \delta(\sigma - \tilde{\sigma}),$$

$$\{\mathcal{P}(\sigma), \mathcal{J}_\pm(\tilde{\sigma})\} = \mathcal{J}_\pm(\sigma)\partial_\sigma \delta(\sigma - \tilde{\sigma}), \quad \{\mathcal{H}_R^{(0)}(\sigma), \mathcal{J}_-(\tilde{\sigma})\} = -\mathcal{J}_-(\sigma)\partial_\sigma \delta(\sigma - \tilde{\sigma}), \tag{3.29}$$

while the commutator of $\mathcal{H}_R^{(0)}$ with $\mathcal{J}_+$ vanishes.[9] Using these, one can easily deduce the commutation relations of the deformed Hamiltonian $\mathcal{H}_R$ with the various other currents, such as

$$\{\mathcal{H}_R(\sigma), \mathcal{H}_R(\tilde{\sigma})\} = -\left(\frac{\mathcal{H}_R(\sigma)}{1 - \lambda\mathcal{J}_+(\sigma) - \frac{\lambda^2}{2}\mathcal{H}_R(\sigma)} + \frac{\mathcal{H}_R(\tilde{\sigma})}{1 - \lambda\mathcal{J}_+(\tilde{\sigma}) - \frac{\lambda^2}{2}\mathcal{H}_R(\tilde{\sigma})}\right)\partial_\sigma \delta(\sigma - \tilde{\sigma}), \tag{3.30}$$

$$\{\mathcal{P}(\sigma), \mathcal{H}_R(\tilde{\sigma})\} = \left(\mathcal{H}_R(\sigma) + \frac{\mathcal{H}_R(\tilde{\sigma})}{1 - \lambda\mathcal{J}_+(\tilde{\sigma}) - \frac{\lambda^2}{2}\mathcal{H}_R(\tilde{\sigma})}\right)\partial_\sigma \delta(\sigma - \tilde{\sigma}). \tag{3.31}$$

In order to arrive at this simple form, we used the identity $\sqrt{(1 - \lambda\mathcal{J}_+)^2 - \lambda^2\mathcal{H}_R^{(0)}} = 1 - \lambda\mathcal{J}_+ - \frac{\lambda^2}{2}\mathcal{H}_R$, as well as the equivalence of distributions spelled out in appendix B.

A more complete list of commutators, whose form is not particularly enlightening, is given in appendix B. An exception is the commutator $\{\mathcal{H}_L(\sigma), \mathcal{H}_L(\tilde{\sigma})\}$ of two left-moving currents, which due to the left conformal invariance of $J\bar{T}$ - deformed CFTs can be shown to take on a standard CFT form

$$\{\mathcal{H}_L(\sigma), \mathcal{H}_L(\tilde{\sigma})\} = (\mathcal{H}_L(\sigma) + \mathcal{H}_L(\tilde{\sigma}))\partial_\sigma \delta(\sigma - \tilde{\sigma}). \tag{3.32}$$

Note that the Poisson bracket (B.5) of the right-moving Hamiltonian with $\phi$ is naturally proportional to a $\delta$ function, thus resolving the potential integration ambiguities that we encountered in the $T\bar{T}$ case.

---

[9]The last two relations follow from $\{\mathcal{H}^{(0)}(\sigma), \mathcal{J}_\pm(\tilde{\sigma})\} = \pm\mathcal{J}_\pm(\sigma)\partial_\sigma \delta(\sigma - \tilde{\sigma})$.

We can now compute the algebra of the symmetry generators (3.12). The details of the calculation are given in appendix B. We find

$$\{Q_f, Q_g\} = Q_{fg'-f'g}, \qquad \{\bar{Q}_{\bar{f}}, \bar{Q}_{\bar{g}}\} = \bar{Q}_{\bar{f}'\bar{g}-\bar{f}\bar{g}'}, \qquad \{Q_f, \bar{Q}_{\bar{f}}\} = 0,$$

which implies that the generators of the diffeomorphisms (3.11) organize themselves into two commuting copies of the Witt algebra. The commutators with the $U(1)$ currents read

$$\{Q_f, P_\chi\} = P_{f\chi'}, \qquad \{P_\chi, P_\eta\} = \frac{1}{2}\int d\sigma \chi \partial_\sigma \eta, \tag{3.33}$$

which represents a $U(1)$ Kac-Moody extension of the left-moving Witt algebra, including the usual Kac-Moody central term with level $k = 1$. The commutator with the right-moving Witt generators vanishes. As for the right-moving $U(1)$ generators, it can be easily shown that they commute with the left-moving Virasoro - Kac-Moody ones. The algebra of these $U(1)$ currents is however not Kac-Moody

$$\{\bar{P}_{\bar{\chi}}, \bar{P}_{\bar{\eta}}\} = -\frac{1}{2}\int d\sigma \, \bar{\chi}\partial_\sigma \bar{\eta} + \frac{\lambda}{2}\bar{P}_{\bar{\chi}\bar{\eta}'-\bar{\chi}'\bar{\eta}}, \tag{3.34}$$

$$\{\bar{Q}_{\bar{f}}, \bar{P}_{\bar{\eta}}\} = -\bar{P}_{\bar{f}\bar{\eta}'} - \frac{\lambda}{2}\bar{Q}_{\bar{f}'\bar{\eta}}.$$

It is not hard to notice that the combination

$$\bar{P}_{\bar{\eta}}^{KM} = \bar{P}_{\bar{\eta}} + \frac{\lambda}{2}\bar{Q}_{\bar{\eta}} \tag{3.35}$$

does have standard Witt - Kac-Moody commutation relations with $\bar{Q}_{\bar{f}}$ and itself, i.e.

$$\{\bar{Q}_{\bar{f}}, \bar{P}_{\bar{\eta}}^{KM}\} = -\bar{P}_{\bar{f}\bar{\eta}'}^{KM}, \qquad \{\bar{P}_{\bar{\chi}}^{KM}, \bar{P}_{\bar{\eta}}^{KM}\} = -\frac{1}{2}\int d\sigma \, \bar{\chi}\partial_\sigma \bar{\eta}. \tag{3.36}$$

Thus, we find two copies of the Witt - Kac-Moody algebra, one of which is completely standard, generated by local transformations, while the second one is field-dependent. Note that if we define the theory on a cylinder then, as before, we should divide the coordinates that appear in the functions labelling the charges by their respective periodicities, i.e. replace $U \to U/R$ and $v \to v/R_v$, where the field-dependent radius of the coordinate $v$ in (3.7) is

$$R_v = R - \lambda\tilde{Q}, \tag{3.37}$$

where $\tilde{Q} = \int d\sigma(\mathcal{J}_+ - \mathcal{J}_-)$ is the winding charge of the field $\phi$. Unlike in the $T\bar{T}$ case, now the field-dependent radius $R_v$ commutes with the energy and charge currents, as well as with the field-dependent coordinate $v$, and thus its inclusion at this final stage does not affect at all the functional charge algebra above.

As we discussed in section section 2, on the cylinder it is natural to label the charges by their Fourier modes. Using the mode functions $f_n(\hat{u}) = e^{in\hat{u}}$ and $\bar{f}_n(\hat{v}) = e^{-in\hat{v}}$, as before, we define

$$L_n = RQ_{f_n}, \qquad \bar{L}_n = R\bar{Q}_{\bar{f}_n}, \qquad K_n = RP_{f_n}, \qquad \bar{K}_n = R\bar{P}_{\bar{f}_n}^{KM}. \tag{3.38}$$

The functional Witt - Kac-Moody algebra we found above then translates into the usual Witt - Kac-Moody algebra for the left-moving Fourier modes

$$\{L_m, L_n\} = -i(m-n)L_{m+n}, \qquad \{L_m, K_n\} = inK_{m+n}, \qquad \{K_m, K_n\} = -iR\frac{m-n}{4}\delta_{m+n}. \tag{3.39}$$

However, the field-dependent radius does enter the right-moving part of the charge algebra, as we obtain

$$\{\bar{L}_m, \bar{L}_n\} = -i\frac{(m-n)}{1-\hat{\lambda}\tilde{Q}}\bar{L}_{m+n}, \qquad \{\bar{L}_m, \bar{K}_n\} = \frac{in}{1-\hat{\lambda}\tilde{Q}}K_{m+n}, \tag{3.40}$$

where $\hat{\lambda} = \lambda/R$ is the dimensionless $J\bar{T}$ coupling on the cylinder. This is our final result for the classical symmetry algebra of a $J\bar{T}$ - deformed free boson. Since this is a classical calculation, we are unable to compute the central extension of the Witt algebra, which is a quantum effect; however, the holographic analysis of [70] indicated, in terms of the *rescaled* charges $RQ_f$ and $R_v\bar{Q}_{\bar{f}}$, it is the same as that of the undeformed CFT.

### 3.3. General $J\bar{T}$-deformed CFTs

So far, we have shown that the $J\bar{T}$ - deformed free boson possesses two sets of Witt $\ltimes$ Kac-Moody symmetries. It is to be expected that general $J\bar{T}$ - deformed CFTs should exhibit a similar enhancement of their global symmetries. To show this, we first need to develop a general Hamiltonian treatment of $J\bar{T}$ - deformed CFTs, along the lines of what was done in [13] for $T\bar{T}$. The goal of this section is to present this general treatment and to construct the conserved charges in a general class of $J\bar{T}$ - deformed CFTs.

We consider a two-dimensional classical field theory with fields $\phi_k$, with $k = 1, \ldots, n$. We assume that the field theory has a $U(1)$ symmetry, which for simplicity will be taken to be a shift symmetry of a scalar field $\phi_1$. The Hamiltonian $\mathcal{H}(\pi_k, \phi_k)$ thus only depends on $\phi_1$ through its spatial derivatives. The $U(1)$ shift current has components

$$J_t^1 = \pi_1, \qquad J_\sigma^1 = \partial_{\phi_1'}\mathcal{H}. \tag{3.41}$$

The components of the topologically conserved current (3.14) are

$$\tilde{J}_t^1 = \phi_1', \qquad \tilde{J}_\sigma^1 = \partial_{\pi_1}\mathcal{H} \tag{3.42}$$

and we assume that in the undeformed CFT, the currents $J^\pm = (J \pm \tilde{J})/2$ are chiral and, respectively, anti-chiral, which will impose certain constraints on $\mathcal{H}^{(0)}$.

We will consider deforming the Hamiltonian by the $J_1\bar{T}$ operator $J_1\bar{T} = J_\sigma^1 T_{tV} - J_t^1 T_{\sigma V}$ constructed from the above current and the components (2.4) of the stress tensor. The flow equation is

$$\partial_\lambda \mathcal{H} = -J_1\bar{T}. \tag{3.43}$$

Introducing, as before, the right-moving energy density $\mathcal{H}_R = (\mathcal{H} - \mathcal{P})/2$, the expression for $J_1\bar{T}$ simplifies to

$$J_1\bar{T} = 2(\pi_1\partial_{\pi_k}\mathcal{H}_R\partial_{\phi_k'}\mathcal{H}_R - \mathcal{H}_R\,\partial_{\phi_1'}\mathcal{H}_R). \tag{3.44}$$

As explained in the previous subsection, the $J\bar{T}$ deformation preserves left conformal invariance, so we expect the $VU$ component of the stress tensor

$$T_{VU} = -\mathcal{H}_R + \pi_k\partial_{\pi_k}\mathcal{H}_R + \phi'^k\partial_{\phi'^k}\mathcal{H}_R + \partial_{\pi_k}\mathcal{H}_R\partial_{\phi'^k}\mathcal{H}_R \tag{3.45}$$

to be zero along the flow. At least in the single-field (i.e., $J\bar{T}$ - deformed free boson) case, this is indeed the case, as $T_{VU}$ can be shown to obey

$$2\partial_\lambda T_{VU} + \pi\partial_\pi\mathcal{H}\,\partial_{\phi'}T_{VU} + \pi\partial_{\phi'}\mathcal{H}\,\partial_\pi T_{VU} - \pi^2\partial_\pi T_{VU} - \mathcal{H}\,\partial_{\phi'}T_{VU} + \pi\,T_{VU} - \partial_{\phi'}\mathcal{H}\,T_{VU} = 0, \tag{3.46}$$

which shows that if $T_{VU}$ starts out being zero, it will stay zero along the flow. We expect an appropriate generalization of this equation to multiple fields to hold as well.

To solve the two equations (3.43) and (3.45), we make the Ansatz

$$\mathcal{H}_R = \frac{1}{\lambda^2} F\left(\lambda^2 \mathcal{H}_R^{(0)}, \lambda \mathcal{J}_+\right) \equiv \frac{1}{\lambda^2} F(x, \alpha), \qquad \mathcal{J}_+ = \frac{\pi_1 + \phi_1'}{2}, \tag{3.47}$$

which is motivated by dimensional analysis, the universality of the deformation, and the fact that we expect, just as in the case of $T\bar{T}$, that the deformed energy formula (3.2) will also apply at the level of the currents, and not only that of the global conserved charges. Since (3.2) only depends on $E_R^{(0)}$ and $Q_+^{(0)}$, we consequently assume that $\mathcal{H}_R$ only depends on $\mathcal{H}_R^{(0)}$ and $\mathcal{J}_+$.

To bring the flow equations into a manageable form, we also need a few properties of the undeformed Hamiltonian. First, since the undeformed theory is a CFT and the Hamiltonian has dimension two, we have

$$\pi_k \partial_{\pi_k} \mathcal{H}_R^{(0)} + \phi_k' \partial_{\phi_k'} \mathcal{H}_R^{(0)} = 2\mathcal{H}_R^{(0)}. \tag{3.48}$$

Since, by assumption, $J^\pm$ are purely (anti-)chiral in the original CFT, $\mathcal{H}_R^{(0)}$ satisfies

$$J_V^{1+} = \frac{1}{2}(\partial_{\pi_1} + \partial_{\phi_1'})\mathcal{H}_R^{(0)} = 0 \tag{3.49}$$

and

$$J_U^{1-} = \frac{1}{2}(\pi_1 - \phi_1') - \frac{1}{2}(\partial_{\pi_1} - \partial_{\phi_1'})\mathcal{H}_R^{(0)} = 0. \tag{3.50}$$

Using these and the Ansatz (3.47), the flow equations can be simplified to

$$-\lambda^2 T_{VU} = F - 2x\partial_x F + x(\partial_x F)^2 - \alpha\partial_\alpha F - \frac{1}{4}(\partial_\alpha F)^2 = 0, \tag{3.51}$$

$$\lambda^3 \left(\partial_\lambda \mathcal{H} + J_1 \bar{T}\right) = -4F + 2(2x - \alpha F)\partial_x F + (2\alpha - F)\partial_\alpha F - 2\lambda\pi_1 \left(x(\partial_x F)^2 - \frac{1}{4}(\partial_\alpha F)^2 - F\partial_x F\right). \tag{3.52}$$

It can be easily checked that

$$F(x, \alpha) = 2(1 - \alpha - \sqrt{(1-\alpha)^2 - x}) \tag{3.53}$$

solves all the above equations. Thus, the Hamiltonian of a rather general $J\bar{T}$ - deformed CFT is given in terms of the undeformed Hamiltonian and the $U(1)$ current via precisely (3.25), where now $\mathcal{H}_R^{(0)}$ is the right-moving Hamiltonian of the undeformed CFT and $\mathcal{J}_+$ is given in (3.47). This expression also agrees with the general deformed energy formula (3.2) (obtained for constant current densities) if we set $k = 2\pi$ in our conventions.

It is furthermore possible to show in full generality that the current

$$K_\alpha = J_\alpha^+ - \frac{\lambda}{2} T_{\alpha V} \tag{3.54}$$

stays chiral along the flow.[10] The chiral component reads

$$K_U = \frac{1}{4}(\pi_1 + \phi_1' + \partial_{\pi_1}\mathcal{H} + \partial_{\phi_1'}\mathcal{H} - 2\lambda T_{UV}) = \frac{1}{2}(\pi_1 + \phi_1' + \lambda\mathcal{H}_R), \tag{3.55}$$

---

[10]Indeed, $K_V = \frac{1}{4}(-\pi_1 - \phi_1' + \partial_{\pi_1}\mathcal{H} + \partial_{\phi_1'}\mathcal{H} - 2\lambda T_{VV}) = \frac{1}{2\lambda}(\partial_\alpha F - \lambda^2 T_{VV}) = 0$ where, using the explicit functional form of $F(x, \alpha)$, it can be shown that $T_{VV} = -\partial_{\pi_k}\mathcal{H}_R \partial_{\phi'^k}\mathcal{H}_R = \frac{1}{\lambda^2}\left[x(\partial_x F)^2 - \frac{1}{4}(\partial_\alpha F)^2\right] = \frac{1}{\lambda^2}\partial_\alpha F$.

which precisely equals the integrand of (3.28), now in the general case. The current $K_U f(U)$ will thus generate an infinite number of conserved $U(1)$ charges, given by (3.28). We note in passing that the combination

$$T_{UU} - K_U^2 = \mathcal{P} + \frac{1}{\lambda^2}(F - (\alpha + F/2)^2) = \mathcal{P} + \mathcal{H}_R^{(0)} - \mathcal{J}_+^2 \tag{3.56}$$

is $\lambda$ - independent. Thus, we see that rather generally, the $J\bar{T}$ deformation induces a type of spectral flow in the left-moving sector.

Next, we would like to check whether in general $J\bar{T}$ - deformed CFTs, the components of the $J^-$ current satisfy the relation (3.19), which would allow us to construct an infinite set of field-dependent, right-moving Kac-Moody charges. In Hamiltonian language, the components of the $J^-$ current read

$$J_U^- = J_V^- + \frac{\pi_1 - \phi_1'}{2}, \qquad J_V^- = \frac{1}{4}(-\pi_1 + \phi_1' - \partial_{\pi_1}\mathcal{H} + \partial_{\phi_1'}\mathcal{H}) = -\frac{1}{2}(\pi_1 - \phi_1')\partial_x F. \tag{3.57}$$

It is trivial to check that $J_U^- T_{VV} + J_V^- T_{UV} = 0$, using the fact that $F$ satisfies $\partial_\alpha F = F\partial_x F$. This implies that, very generally, the ratio of the $J^-$ components is the same as that of the right-moving stress tensor, allowing us to construct an infinite set of field-dependent right-moving $U(1)$ symmetries.

The conserved charges are given by the same formulae (3.27) and (3.28) as in the free boson case, but now in terms of the general initial Hamiltonian density and currents. The field-dependent coordinate[11] is $v = V - \lambda\phi_1$. Since the commutation relations (3.29) of the undeformed generators hold in any CFT and the formula (3.25) for the deformed Hamiltonian is universal, it is clear that the commutation relations of the conserved charges are exactly the same as in the free boson case, namely two commuting copies of the Witt - Kac-Moody algebra, provided we redefine the right-moving $U(1)$ generator as in (3.35).

# 4. $JT_a$-deformed CFTs

As already emphasized in the previous section, the $J\bar{T}$ deformation has the nice property that it preserves half of the conformal symmetries of the original CFT. However, given that the main message of this article is that all Smirnov–Zamolodchikov-type deformations of two-dimensional CFTs preserve all the infinite-dimensional symmetries of the original CFT in a particular modified form, it is interesting to analyse these more general deformations, too.

Thus, in this section we study irrelevant deformations constructed from the components of a $U(1)$ current and the generator $T_{\alpha a}$ of translations in some chosen direction $\hat{x}^a$, defined via

$$\partial_{\lambda^a} S = \int d^2x \sqrt{\gamma}\, \epsilon^{\alpha\beta} J_\alpha T_{\beta a}, \tag{4.1}$$

---

[11] In the Hamiltonian formalism, the derivatives of the field-dependent coordinate $v$ are given by

$$\partial_U v = -\frac{\lambda}{2}(\partial_{\pi_1}\mathcal{H} + \phi_1'), \qquad \partial_V v = 1 + \frac{\lambda}{2}(\partial_{\pi_1}\mathcal{H} - \phi_1') \tag{3.58}$$

and it can be easily checked that they satisfy

$$\partial_U v\, T_{VV} + \partial_V v\, T_{UV} = \frac{1}{\lambda^2}(\partial_\alpha F - F - \alpha F\partial_x F - \frac{1}{2}F\partial_\alpha F) + \frac{\phi_1'}{\lambda}(F\partial_x F - \partial_\alpha F) = 0. \tag{3.59}$$

It is interesting to note that this identity also holds if $\pi_1 \leftrightarrow \phi_1'$, which interchanges $J$ with $\tilde{J}$.



where we work in conventions where $\epsilon_{t\sigma} = \epsilon^{\sigma t} = 1$. We will in fact distinguish between two different types of deformations, denoted $JT_a$ and $\tilde{J}T_a$, the first constructed from the usual Noether current $J$ that is only conserved on-shell, and the other constructed from the topological current $\tilde{J} = \star d\phi$, whose conservation is an off-shell identity.

Despite the fact that $JT_a$ deformations generally break both conformal and Lorentz invariance, the finite-size spectrum of such theories could be derived in [56] using coupling to background fields. The formula for the deformed energy in finite size reads

$$E(\lambda_a, R) = \frac{\lambda_\sigma}{\lambda_t} P^{(0)} - \frac{\lambda \cdot Q}{\lambda_t^2} + \frac{R}{\lambda_t^2} \left( 1 - \sqrt{\left(1 - \frac{\lambda \cdot Q}{R}\right)^2 - \frac{2\lambda_t^2 E^{(0)}}{R} + \frac{2\lambda_t \lambda_\sigma P^{(0)}}{R} - \frac{2\lambda_t(\lambda_\sigma^2 - \lambda_t^2)}{R^2} P^{(0)} Q} \right),$$
(4.2)

where $\lambda_t, \lambda_\sigma$ are the time and, respectively, space components of the deformation vector $\lambda_a$,

$$\lambda \cdot Q \equiv \lambda_\sigma Q + \lambda_t \tilde{Q},$$
(4.3)

$R$ is the circumference of the cylinder and $E^{(0)}, P^{(0)}, Q = \int J$ and $\tilde{Q} = \int \tilde{J}$ are the energy, momentum and conserved charges of the corresponding state in the undeformed CFT.[12] The formula for the energy spectrum of a $\tilde{J}T_a$ - deformed CFT is simply obtained via the replacement $Q \leftrightarrow \tilde{Q}$.

We start by studying the $JT_a$ and $\tilde{J}T_a$ - deformed free boson systems, whose Lagrangians can be built explicitly, and construct the infinite-dimensional pseudo-conformal and Kac-Moody symmetries there first. Interestingly, while the Lagrangians of the two theories look rather different, their Hamiltonians are related in a very simple manner. We then present the deformed Hamiltonian density of a general $JT_a$ - deformed CFT and show that the conserved charges charges organise themselves yet again into two commuting copies of the Witt - Kac-Moody algebra.

## 4.1. $JT_a$-deformed free boson

In this subsection, we work out the $JT_a$-deformed free boson action and symmetries. The general form of the action is

$$S = \int dt \, d\sigma \, \mathcal{L}(\dot{\phi}, \phi').$$
(4.4)

The canonical stress tensor is given by

$$T_{\alpha\beta} = \gamma_{\alpha\beta} \mathcal{L} - \gamma_{\alpha\gamma} \frac{\partial \mathcal{L}}{\partial(\partial_\gamma \phi)} \partial_\beta \phi$$
(4.5)

and its components are, explicitly

$$T_{tt} = \frac{\partial \mathcal{L}}{\partial \dot{\phi}} \dot{\phi} - \mathcal{L}, \quad T_{t\sigma} = \frac{\partial \mathcal{L}}{\partial \dot{\phi}} \phi', \quad T_{\sigma t} = -\frac{\partial \mathcal{L}}{\partial \phi'} \dot{\phi}, \quad T_{\sigma\sigma} = \mathcal{L} - \frac{\partial \mathcal{L}}{\partial \phi'} \phi'.$$
(4.6)

The components of the shift current are

$$J^\alpha = -\frac{\partial \mathcal{L}}{\partial(\partial_\alpha \phi)} \quad \Rightarrow \quad J_t = \frac{\partial \mathcal{L}}{\partial \dot{\phi}}, \quad J_\sigma = -\frac{\partial \mathcal{L}}{\partial \phi'}.$$
(4.7)

---

[12]Note that $\lambda_\sigma = -\alpha_{JT_1}$ in [56], and the sign of $P^{(0)}$ is flipped with respect to this reference; also $\lambda_\sigma = -\mu_\phi$ with respect to the conventions of [57].

The $JT_a$ flow equations take the form

$$\frac{\partial \mathcal{L}}{\partial \lambda^a} = (JT_a) = \epsilon^{\alpha\beta} J_\alpha T_{\beta a}, \quad \text{with} \quad JT_t = \mathcal{L}\frac{\partial \mathcal{L}}{\partial \phi'}, \quad JT_\sigma = -\mathcal{L}\frac{\partial \mathcal{L}}{\partial \dot{\phi}}, \tag{4.8}$$

which, as explained in [57], are satisfied separately for $\lambda_\sigma$ and $\lambda_t$. These equations are of Burger's type, with solution ($\epsilon_{t\sigma} = 1$)

$$\mathcal{L}(\phi, \phi') = \mathcal{L}_0(\dot{\phi} - \lambda^\sigma \mathcal{L}, \phi' + \lambda^t \mathcal{L}) = \mathcal{L}_0(\partial_a \phi - \epsilon_{ab}\lambda^b \mathcal{L}), \tag{4.9}$$

where $\mathcal{L}_0$ is the undeformed Lagrangian. Taking $\mathcal{L}_0 = \frac{1}{2}(\dot{\phi}^2 - \phi'^2)$, the deformed Lagrangian satisfies

$$\lambda^2 \mathcal{L}^2 - 2\mathcal{L}(1 - \epsilon^{ab}\partial_\alpha \phi \lambda_b) - \eta^{ab}\partial_a \phi \partial_b \phi = 0, \tag{4.10}$$

where $\lambda^2 = \lambda_a \lambda^a$. In the case of $J\bar{T}$, $\lambda^2 = 0$, and we obtain the solution of the previous section, with $\lambda_{J\bar{T}} = 2\lambda_\sigma = 2\lambda_t$. Concentrating on $\lambda^2 \neq 0$, we find

$$\mathcal{L}_{JT_a} = \frac{1}{\lambda^2}\left(1 - \epsilon^{ab}\partial_a \phi \lambda_b - \sqrt{1 - 2\epsilon^{ab}\partial_a \phi \lambda_b + (\lambda^a \partial_a \phi)^2}\right). \tag{4.11}$$

Passing to null coordinates $U, V = \sigma \pm t$, it is not hard to check that[13]

$$\lambda^U J_V T_{UU} + (1 - \lambda^U J_U)T_{VU} = 0, \quad (1 + \lambda^V J_V)T_{UV} - \lambda^V J_U T_{VV} = 0. \tag{4.12}$$

According to our general discussion in the introduction, this implies that if we introduce the field-dependent coordinates

$$u = U - \lambda^U \varphi, \quad v = V - \lambda^V \varphi, \tag{4.13}$$

where $\varphi$ is a scalar in term of which $J = \star d\varphi$ (i.e., $J_U = \partial_U \varphi$ and $J_V = -\partial_V \varphi$), then we obtain two infinite sets of field-dependent symmetries, with conserved charges given by

$$Q_f = \int d\sigma f(u)(T_{UU} - T_{VU}), \quad \bar{Q}_{\bar{f}} = \int d\sigma \bar{f}(v)(T_{VV} - T_{UV}), \tag{4.14}$$

where $f(u), \bar{f}(v)$ are arbitrary functions of the corresponding field-dependent coordinates (4.13). If the theory is defined on a compact space, then charge conservation requires the functions $f(u), \bar{f}(v)$ to be periodic and it is natural to normalize the field-dependent coordinates by the corresponding field-dependent radii $R_u = R - \lambda^U Q$ and $R_v = R - \lambda^V Q$.

The theory will similarly possess an infinite set of of conserved $U(1)$ currents, which are analogous to $J^\pm$ in the undeformed CFT. The seed currents are

$$K_\alpha \equiv \frac{1}{2}(J_\alpha + \tilde{J}_\alpha - \lambda^V T_{\alpha V}), \quad \bar{K}_\alpha \equiv \frac{1}{2}(J_\alpha - \tilde{J}_\alpha + \lambda^U T_{\alpha U}) \tag{4.15}$$

and can be shown to satisfy $K_U/K_V = T_{UU}/T_{VU}$ and respectively $\bar{K}_U/\bar{K}_V = T_{UV}/T_{VV}$. Thus $\chi(u)K_\alpha$ and $\bar{\chi}(v)\bar{K}_\alpha$ will be conserved for any functions $\chi(u), \bar{\chi}(v)$ of the field-dependent coordinates (4.13) associated with the $JT_a$ deformation. Notice that one can add an arbitrary multiple of $T_{\alpha U}$ to $K_\alpha$ and an arbitrary multiple of $T_{\alpha V}$ to $\bar{K}_\alpha$ without affecting their conservation. As we will see shortly, an appropriate linear combination of these will generate two infinite sets of Kac-Moody symmetries.

The relation between the auxiliary scalar $\varphi$ that enters the field-dependent coordinates and the 'fundamental' scalar $\phi$ is rather non-local; yet, the conserved charges depend explicitly on it. To have conserved charges that only depend on the local field, it may be more natural to

---

[13]Remember that $\lambda^U = 2\lambda_V = \lambda_\sigma - \lambda_t$, $\lambda^V = 2\lambda_U = \lambda_\sigma + \lambda_t$.

study the $\tilde{J}T_a$ deformation instead, for which we expect the shift of the coordinates to just be proportional to the fundamental boson.

It can also be checked that, unlike in the case of $J\bar{T}$, where we could choose (as remarked upon in footnote 11) the field-dependent coordinates to be either given by the analogue of (4.13), or that of (4.21), with the latter being the more natural choice, in $JT_a$ the only possible field-dependent coordinate is (4.13).

## 4.2. $\tilde{J}T_a$ - deformed free boson

The $\tilde{J}T_a$ deformation is defined in the same way as the $JT_a$ one (4.1), except that the components of $J$ are now replaced by those of the current $\tilde{J}$

$$\tilde{J}_t = \phi', \quad \tilde{J}_\sigma = \dot{\phi}. \tag{4.16}$$

The flow equations read

$$\frac{\partial \mathcal{L}}{\partial \lambda_\sigma} = \phi'(\dot{\phi}\,\partial_{\dot{\phi}}\mathcal{L} + \phi'\partial_{\phi'}\mathcal{L} - \mathcal{L}), \quad \frac{\partial \mathcal{L}}{\partial \lambda_t} = -\dot{\phi}(\dot{\phi}\,\partial_{\dot{\phi}}\mathcal{L} + \phi'\partial_{\phi'}\mathcal{L} - \mathcal{L}). \tag{4.17}$$

From this, we immediately deduce that $\dot{\phi}\,\partial_{\lambda_\sigma}\mathcal{L} + \phi'\partial_{\lambda_t}\mathcal{L} = 0$. Dimensional analysis suggests the Ansatz

$$\mathcal{L} = \frac{1}{\lambda^2}F(x,y), \quad x = \lambda^a\partial_a\phi = \lambda_\sigma\phi' - \lambda_t\dot{\phi}, \quad y = \lambda_a\epsilon^{ab}\partial_b\phi = \lambda_\sigma\dot{\phi} - \lambda_t\phi'. \tag{4.18}$$

The equation above reduces to $2yF = (y^2 - x^2)\partial_y F$, implying that $F(x,y) = (y^2 - x^2)F_0(x)$. To fix $F_0$, we plug this Ansatz into the flow equations (4.17), obtaining

$$F(x,y) = \frac{y^2 - x^2}{2(1-x)}, \tag{4.19}$$

where the overall coefficient is fixed by matching to the free boson as $\lambda_{\sigma,t} \to 0$. The Lagrangian of the $\tilde{J}T_a$ - deformed free boson thus takes a very simple form

$$\mathcal{L}_{\tilde{J}T_a} = \frac{\dot{\phi}^2 - \phi'^2}{2(1 - \lambda^a\partial_a\phi)}, \tag{4.20}$$

which is a straightforward generalization of the $J\bar{T}$ Lagrangian (3.4) - (3.5). This action is equivalent to that of a free boson via the field-dependent coordinate transformation

$$u^a = U^a - \lambda^a\phi. \tag{4.21}$$

It is interesting that the Lagrangians for the $JT_a$ and $\tilde{J}T_a$ - deformed free boson look so different. However, we expect the associated Hamiltonians to be very similar, as we will soon show.

Passing to the null coordinates $U,V$, the components of the shift current read

$$J_U = \frac{\partial_U\phi(1 - \lambda^U\partial_U\phi)}{(1 - \lambda^a\partial_a\phi)^2}, \quad J_V = \frac{\partial_V\phi(1 - \lambda^V\partial_V\phi)}{(1 - \lambda^a\partial_a\phi)^2} \tag{4.22}$$

and those of the stress tensor

$$T_{UU} = J_U\partial_U\phi, \quad T_{VV} = J_V\partial_V\phi, \quad T_{VU} = \frac{\partial_U\phi\,\partial_V\phi\,\lambda^U\partial_U\phi}{(1 - \lambda^a\partial_a\phi)^2}, \quad T_{UV} = J_U\partial_V\phi - \mathcal{L}. \tag{4.23}$$

It can be easily checked that the field-dependent coordinates (4.21) proposed above satisfy (1.2), implying yet again the existence of two infinite sets of pseudo-conformal symmetries. In contrast with $JT_a$, now the generators of the symmetries are entirely constructed from the fundamental scalar $\phi$.

Let us now turn to the currents. It is not hard to check that the combinations

$$\mathcal{K}_\alpha \equiv \frac{1}{2}(J_\alpha + \tilde{J}_\alpha - \lambda^V T_{\alpha V}), \qquad \bar{\mathcal{K}}_\alpha \equiv \frac{1}{2}(J_\alpha - \tilde{J}_\alpha - \lambda^U T_{\alpha U}) \tag{4.24}$$

satisfy $\mathcal{K}_U/\mathcal{K}_V = T_{UU}/T_{VU}$ and respectively $\bar{\mathcal{K}}_U/\bar{\mathcal{K}}_V = T_{UV}/T_{VV}$, and thus $\chi(u)K_\alpha$ and $\bar{\chi}(v)\bar{K}_\alpha$ will be conserved for any functions $\chi, \bar{\chi}$ of the field-dependent coordinates (4.21). These will generate two sets of Kac-Moody currents. As before, one can add an arbitrary multiple of $T_{\alpha U}$ to $\mathcal{K}_\alpha$ and of $T_{\alpha V}$ to $\bar{\mathcal{K}}_\alpha$, without affecting this conclusion.

## 4.3. Hamiltonian analysis and general $JT_a$ - deformed CFTs

We now move on to the Hamiltonian description of the $JT_a$ and $\tilde{J}T_a$ - deformed free boson. In the $JT_a$ case, the Lagrangian is given by (4.11), and the momentum conjugate to $\phi$ is

$$\pi^{(JT_a)} = \frac{\partial \mathcal{L}}{\partial \dot{\phi}} = \frac{\lambda_\sigma}{\lambda^2}\left(1 - \frac{1}{\sqrt{1 - 2\epsilon^{ab}\partial_a\phi\lambda_b + (\lambda^a\partial_a\phi)^2}}\right) + \frac{\lambda_t}{\lambda^2}\frac{\lambda^a\partial_a\phi}{\sqrt{1 - 2\epsilon^{ab}\partial_a\phi\lambda_b + (\lambda^a\partial_a\phi)^2}}. \tag{4.25}$$

The equation for $\dot{\phi}$ in terms of $\phi'$ and $\pi$ is quadratic, with solution

$$\dot{\phi} = -\frac{\lambda_\sigma(1 - \lambda_t\phi')}{\lambda_t^2} + \frac{\lambda_\sigma - \pi\lambda^2}{\lambda_t^2}\sqrt{\frac{1 - 2\lambda_t\phi'}{1 - 2\pi\lambda_\sigma + \pi^2\lambda^2}}, \tag{4.26}$$

where we assumed that $\lambda_\sigma > \pi\lambda^2$ and $\lambda^2 > 0$. Plugging into the Hamiltonian density, we find

$$\mathcal{H}_{JT_a}^{f.b.} = \frac{(1 - \pi\lambda_\sigma)(1 - \phi'\lambda_t)}{\lambda_t^2} - \frac{\sqrt{(1 - 2\phi'\lambda_t)(1 - 2\pi\lambda_\sigma + \pi^2\lambda^2)}}{\lambda_t^2}. \tag{4.27}$$

This Hamiltonian density can be rewritten in the suggestive form

$$\mathcal{H}_{JT_a} = \frac{\lambda_\sigma}{\lambda_t}\mathcal{P} - \frac{\lambda \cdot \mathcal{J}}{\lambda_t^2} + \frac{1}{\lambda_t^2}\left(1 - \sqrt{(1 - \lambda \cdot \mathcal{J})^2 + 2\lambda_t(\mathcal{P}\lambda_\sigma - \lambda_t\mathcal{H}^{(0)} - \lambda^2\mathcal{P}(\mathcal{J}_+ + \mathcal{J}_-))}\right), \tag{4.28}$$

where

$$\lambda \cdot \mathcal{J} = \lambda_\sigma(\mathcal{J}_+ + \mathcal{J}_-) + \lambda_t(\mathcal{J}_+ - \mathcal{J}_-) = 2\lambda_U\mathcal{J}_+ + 2\lambda_V\mathcal{J}_- \tag{4.29}$$

in analogy with (4.3). Remember that for the free boson, $\mathcal{P} = \pi\phi'$ and $\mathcal{H}^{(0)} = (\pi^2 + \phi'^2)/2$. It is now clear that a uniform density state of the deformed system in finite size will have an energy given by (4.2).

It is interesting to also derive the Hamiltonian for the $\tilde{J}T_a$ - deformed free boson, for which the Lagrangian takes the very different form (4.20). The conjugate momentum is

$$\pi^{(\tilde{J}T_a)} = \frac{\dot{\phi}(1 - \lambda_\sigma\phi') + \lambda_t(\dot{\phi}^2 + \phi'^2)/2}{(1 + \lambda_t\dot{\phi} - \lambda_\sigma\phi')^2} \tag{4.30}$$

and so

$$\dot{\phi} = -\frac{1 - \lambda\phi'}{\lambda_t} + \frac{1}{\lambda_t}\sqrt{\frac{1 - 2\lambda_\sigma\phi' + \lambda^2\phi'^2}{1 - 2\lambda_t\pi}}. \tag{4.31}$$

The Hamiltonian density of the $\tilde{J}T_a$ - deformed free boson then reads

$$\mathcal{H}^{f.b.}_{\tilde{J}T_a} = \frac{1}{\lambda_t^2}\left((1-\lambda_\sigma\phi')(1-\lambda_t\pi) - \sqrt{(1-2\lambda_t\pi)(1-2\lambda_\sigma\phi'+\lambda^2\phi'^2)}\right). \qquad (4.32)$$

Note this is related to $\mathcal{H}_{JT_a}$ by the simple interchange $\pi \leftrightarrow \phi'$, as suggested by the energy formula given in [56]. Therefore, we no longer need to discuss the $\tilde{J}T_a$ - deformation, given that in the Hamiltonian formalism, it is identical to the $JT_a$ deformation, up to the interchange $\pi \leftrightarrow \phi'$.

The Hamiltonian density of a $JT_a$ - deformed CFT satisfies the flow equation

$$\partial_{\lambda^a}\mathcal{H} = -(JT)_a = -\epsilon^{\alpha\beta}J_a T_{\beta a}, \qquad (4.33)$$

which should hold for general $JT_a$ - deformed CFTs, at least of the kind described in section, not only the free boson. Using the expression (2.4) for the stress tensor components and (3.41) for those of the $U(1)$ current, the flow equations take the specific form

$$\partial_{\lambda_t}\mathcal{H} = (JT)_t = -\pi_1\partial_{\pi_k}\mathcal{H}\partial_{\phi'^k}\mathcal{H} + \mathcal{H}\partial_{\phi'_1}\mathcal{H}, \qquad (4.34)$$

$$\partial_{\lambda_\sigma}\mathcal{H} = -(JT)_\sigma = -\mathcal{P}\,\partial_{\phi'_1}\mathcal{H} + \pi_1(\pi_k\partial_{\pi_k}\mathcal{H} + \phi'^k\partial_{\phi'^k}\mathcal{H} - \mathcal{H}). \qquad (4.35)$$

While we will not attempt to solve this equation, the experience of [13] and of the $J\bar{T}$ deformation makes it rather clear that the solution should simply be given by the same formula as that for the deformed energy, but now with the constant densities replaced by general currents. Indeed, plugging the general Hamiltonian (4.28), with $\mathcal{H}^{(0)}, \mathcal{P}$ and $\mathcal{J}_\pm$ replaced by the Hamiltonian, momentum and charge densities of an arbitrary seed CFT, into (4.35) and making use of only the general constraints that the Hamiltonian $\mathcal{H}^{(0)}$ of the original CFT satisfies, such as the conformality constraint (2.5) and the chirality constraints (3.49) and (3.50) on the CFT currents,[14] we find that it solves the flow equations. Thus, (4.28) represents the general expression for the Hamiltonian density of a $JT_a$ - deformed CFT.

As noted above, everything we said so far also holds for the $\tilde{J}T_a$ deformation, upon the simple interchange $\pi_1 \leftrightarrow \phi'_1$, or $\mathcal{J}_- \to -\mathcal{J}_-$.

## 4.4. Conserved charges and their algebra

The conserved charges associated with the pseudo-conformal symmetries and their $U(1)$ counterparts (4.15) are given by

$$Q_f = \int d\sigma f(u)\mathcal{H}_L(\sigma), \qquad \bar{Q}_{\bar{f}} = \int d\sigma \bar{f}(v)\mathcal{H}_R(\sigma), \qquad (4.36)$$

$$P_\chi = \int d\sigma \chi(u)(\mathcal{J}_+ + \lambda_U \mathcal{H}_R), \qquad \bar{P}_{\bar{\chi}} = \int d\sigma \bar{\chi}(v)(\mathcal{J}_- + \lambda_V \mathcal{H}_L), \qquad (4.37)$$

where, as before, $\mathcal{H}_{L,R} = (\mathcal{H} \pm \mathcal{P})/2$, were now $\mathcal{H}$ is given by the general expression (4.28), and $u, v$ are given in (4.13).

The commutation relations can be evaluated as before starting from those of $\mathcal{H}^{(0)}, \mathcal{P}, \mathcal{J}_\pm$. The commutation relations of the auxiliary field $\varphi$ that enters the definition of the field-dependent coordinates are determined from those of the currents, via

$$\varphi' = J_t = \mathcal{J}_+ + \mathcal{J}_- \qquad (4.38)$$

---

[14]In terms of $\mathcal{H}^{(0)}$, these amount to $\partial_{\pi_1}\mathcal{H}^{(0)} = \pi_1$, $\partial_{\phi'_1}\mathcal{H}^{(0)} = \phi'_1$.

and integrating with respect to $\sigma$.

The details of the calculation are unilluminating, and the steps followed are the same as in the analogous computation for $J\bar{T}$.[15] The charge algebra splits again into two commuting copies of the Witt- $U(1)$ current algebra, given by

$$\{Q_f, Q_g\} = Q_{fg'-f'g}, \qquad \{Q_f, P_\chi\} = P_{f\chi'} - \lambda_V Q_{\chi f'}, \qquad \{P_\chi, P_\eta\} = \frac{1}{2}\int d\sigma\,\chi\,\partial_\sigma\eta + \lambda_V P_{\chi\eta'-\eta\chi'} \tag{4.39}$$

and

$$\{\bar{Q}_{\bar{f}}, \bar{Q}_{\bar{g}}\} = \bar{Q}_{\bar{f}'\bar{g}-\bar{g}'\bar{f}}, \quad \{\bar{Q}_{\bar{f}}, \bar{P}_{\bar{\chi}}\} = -\bar{P}_{\bar{f}\bar{\chi}'} + \lambda_U\bar{Q}_{\bar{f}'\bar{\chi}}, \quad \{\bar{P}_{\bar{\chi}}, \bar{P}_{\bar{\eta}}\} = -\frac{1}{2}\int d\sigma\,\bar{\chi}\,\partial_\sigma\bar{\eta} + \lambda_U\bar{P}_{\bar{\chi}'\bar{\eta}-\bar{\eta}'\bar{\chi}}, \tag{4.40}$$

where all the commutators involving one left- and one right-moving generator vanish. The algebra can be put into the standard Witt - Kac-Moody form via the redefinition

$$P_\chi^{KM} = P_\chi - \lambda_V Q_\chi, \qquad \bar{P}_{\bar{\chi}}^{KM} = \bar{P}_{\bar{\chi}} - \lambda_U\bar{Q}_{\bar{\chi}}. \tag{4.41}$$

If we are on compact space, the field-dependent coordinates $u, v$ should be divided by the corresponding field-dependent radii

$$R_u = R - \lambda^U Q, \qquad R_v = R - \lambda^V Q. \tag{4.42}$$

As in the $J\bar{T}$ case, these radii do not contribute to the Poisson brackets of the charges, because they commute with all the energy and charge currents, as well as with the field-dependent coordinates. As before, when expressing the above algebra in terms of Fourier modes, the factors $R/R_u$ and $R/R_v$ will appear explicitly on the right-hand side of the left- and, respectively, right-moving commutators.

# 5. Discussion

In this article, we studied the symmetries of classical $T\bar{T}, J\bar{T}$ and $JT_a$ - deformed CFTs. We showed that each deformed theory possesses an infinite number of conserved charges, which are in one-to-one correspondence with the initial conformal and, if applicable, affine $U(1)$ symmetries of the undeformed CFT. The manner in which the symmetries of the initial CFT are deformed is rigidly dictated by the field-dependent coordinates associated to each of these deformations, i.e. the coordinates in terms of which the classical dynamics of the deformed theory reduces to that of the original CFT. The charge algebra that we find in all cases consists of two commuting copies of the functional Witt - Kac-Moody algebra, i.e., it is functionally the same as that of the undeformed CFT. While these conclusions were already present in the holographic analyses of [28] and [70], this work sharpens and complements them in a formalism that is more straightforwardly applicable to the quantum case.

The symmetries of the deformed CFTs can be analysed either on the plane, or on the cylinder. In this article, we mostly concentrated on the compact case, as it presents a number of

---

[15]As in $J\bar{T}$ (but unlike in $T\bar{T}$), a great simplification comes from the fact that the commutator of the Hamiltonian density with $\tilde{\varphi}'$ is trivial to integrate, as in (B.5), since the only $\tilde{\sigma}$ dependence is in the $\delta$ function. The resulting commutator with $\tilde{\phi}$ is proportional to a delta function. Since this is a local result, it is natural to set the possible integration function to zero, thus fixing the ambiguities that plagued the $T\bar{T}$ calculation. Another simplification that allows us to easily compute the charge Poisson brackets is that, as in (B.11) for $J\bar{T}$, the $\tilde{\sigma}$ derivative of the Poisson bracket of any two currents such as for example $\partial_{\tilde{\sigma}}(\{\mathcal{H}_L, \tilde{\mathcal{H}}_L\})_{\tilde{\sigma}=\sigma}$, when evaluated at $\tilde{\sigma} = \sigma$, is a total $\sigma$ derivative, implying that the term can be easily integrated by parts. As before, the constant term in this integration by parts needs to be judiciously chosen, in order to obtain a meaningful algebra.

subtleties that disappear in the infinite-radius limit. These subtleties are due to the fact that the arbitrary functions of the field-dependent coordinates that label the conserved charges need to be periodic in the compact case; this requirement is fulfilled by dividing their argument by the field-dependent radius of the respective coordinate. This has several effects: i) the field-dependent radius appears explicitly in the charge algebra when written in terms of Fourier modes and thus, the algebra we obtain is not quite Witt at the level of the mode functions; and ii) in the $T\bar{T}$ case, the field-dependent radius contributes non-trivially to the Poisson brackets of the charges. These contributions are not conserved, but they can be cancelled at the cost of including certain winding terms in the Poisson brackets of the field-dependent coordinates.

These additional winding terms are allowed due to certain ambiguities in the Poisson brackets of the $T\bar{T}$ field-dependent coordinates, which our analysis did not fix. In fact, we found two holomorphic and two anti-holomorphic functions' worth of freedom entering these Poisson brackets, resulting in a corresponding ambiguity in the $T\bar{T}$ charge algebra. In contrast, the analogous would-be ambiguities for the $J\bar{T}$ and $JT_a$ deformations are naturally resolved by the fact that the current can always be written in terms of a free boson, whose commutation relations are known. We therefore expect that a better understanding of the physical interpretation of the fields $\int d\sigma(\mathcal{H} \pm \mathcal{P})$, which appear in the definition of the $T\bar{T}$ field-dependent coordinates, would allow us to unambiguously fix this charge algebra, too. In particular, it should explain the origin of the winding terms in the Poisson brackets (2.40) that are apparently necessary for charge conservation. As we currently lack such an understanding, we have presented the more general result.

Our analysis has also clarified the role of the quantities $\mathcal{L}, \bar{\mathcal{L}}$ in a $T\bar{T}$ - deformed CFT, which behave as the stress tensor components of an auxiliary CFT living in the field-dependent coordinates $u, v$, from the point of view of the Hamiltonian formalism. Namely, we showed that the equal-time Poisson brackets of these quantities match, to a convincing extent, the unequal-time Poisson brackets of the stress tensor components in the would-be CFT. This observation is however not very useful in computing the charge algebra, as one would still need to separately evaluate the contributions coming from the Poisson brackets of the field-dependent coordinates, although it could provide an alternate and likely instructive way of performing the calculation.

Looking forward, the thorough classical analysis we presented herein is the first step towards building the quantum symmetries of the system. Nevertheless, a lot more work is required to determine if the infinitely-extended symmetries actually survive quantization, and to understand how to appropriately define the charge operators. Not only will this process involve resolving the operator ordering ambiguities and finding the central terms that likely appear, but a crucial step will be to determine how the symmetry generators act on the states of the system. It will be particularly interesting to find how the representations of this rather peculiar kind of field-dependent symmetries are organised.

An important indication that the action of these symmetries on the states will be subtle, is the fact that the Witt algebra we found is not compatible with the finite-size spectrum of the deformed theories: indeed, while the former predicts the usual integer spacing between descendant states obtained by acting with the raising operators $L_{-m}$ and $\bar{L}_{-n}$, the latter predicts a spacing given by the difference of two square roots. This tension persists upon including in the algebra the corrections due to the field-dependent radius. One possible explanation for this disagreement is suggested by a subtlety that can already be seen in the $J\bar{T}$ and $JT_a$ - deformations: the field-dependent symmetry generators may continuously deform the winding and momentum of the states, which in the quantum theory are supposed to obey a quantization condition. If confirmed, this would indicate that the charges we constructed do not act within the Hilbert space of physical states of the deformed CFTs placed on a cylinder. This may not be a very big problem in itself, as these theories are already ill-defined on the cylinder, due

to the appearance of imaginary energies. It is also conceivable that a slight modification of the symmetry generators makes the algebra compatible with the spectrum on compact space, resolving this conflict.

Whereas the action of the symmetries we found raises questions in compact space, these issues are avoided on the plane, where no such quantization conditions are required. Moreover, in this case the winding terms in the $T\bar{T}$ Poisson brackets disappear and the charge algebra is exactly Witt. Since the non-compact spectrum is not affected by the deformations we considered, in this case we need to understand how the charge operators act on more complicated observables, such as local operators.

If the infinite set of symmetries does survive quantization and their action is understood for $T\bar{T}, J\bar{T}$ and $JT_a$ - deformed CFTs, it would be of great interest to understand their implications in more general settings. For example, the single-trace versions of the $T\bar{T}$ and $J\bar{T}$ deformations share many properties with the Smirnov–Zamolodchikov double-trace ones. Given the crucial importance of the dynamical coordinates in the existence of the infinite symmetries we found, it would be interesting to see whether they contain an analogue of the field-dependent coordinates, or perhaps a more general mechanism for the implementation of the symmetry. Since these deformations are dual to string theory in non-asymptotically AdS spacetimes, this may provide a powerful tool for developing the holographic dictionary in such spacetimes.

# Acknowledgements

The authors would like to thank Eva Llabres for collaboration in the early stages of this project. This research was supported in part by the ERC Starting Grant 679278 Emergent-BH and the Swedish Research Council grant number 2015-05333. RM also received support from NSF grant PHY-19-14412.

# A. Details of the Poisson bracket algebra in $T\bar{T}$-deformed CFTs

## A.1. Constraints from charge conservation

As explained in the main text, the Poisson brackets of the field-dependent coordinates are obtained by integrating (2.29) with respect to $\sigma, \tilde{\sigma}$. This results in a certain ambiguity in the Poisson brackets, which we parametrized in terms of four functions: $A, B, D_{\pm}$.

These functions are not entirely arbitrary; in particular, they should be compatible with charge conservation. In Hamiltonian language, the time derivative of the charges (2.19), (2.20) is given by

$$\frac{d}{dt}Q_f = \partial_t Q_f - \{H, Q_f\} = \frac{1}{2}\int d\sigma\left[\frac{f'(\hat{u})}{R_u}(\mathcal{H}+\mathcal{P}) - \frac{f'(\hat{u})}{R_u}(\mathcal{H}+\mathcal{P})\{H,u\} - f(\hat{u})\{H,\mathcal{H}+\mathcal{P}\}\right],$$
(A.1)

where $H$ is the full Hamiltonian. Using (2.28) and (2.32), we find

$$\{H, \mathcal{H}+\mathcal{P}\} = -\partial_\sigma\left((\mathcal{H}+\mathcal{P})\frac{1+2\mu\mathcal{P}}{1+2\mu\mathcal{H}}\right),$$
(A.2)

$$\{H, u\} = \frac{\mu(\mathcal{H}-\mathcal{P})}{1+2\mu\mathcal{H}}(1-2\mu\mathcal{P}) - \mu(w_{D_-} - G_-(0)) - \mu^2(w_B - F(0)).$$
(A.3)

Integrating by parts the last term in (A.1), we find

$$\frac{d}{dt}Q_f = \frac{Q_{f'}}{R_u}[G_-(0) - w_{D_-} + \mu(F(0) - w_B)]. \tag{A.4}$$

Thus, the left-moving charges are conserved only if the right-hand side of (A.4) vanishes. A similar analysis can be performed for the right-moving charges, finding that (2.19), (2.20) are only conserved if the constraint (2.34) holds. The second relation follows from conservation of $\bar{Q}_f$.

## A.2. Time evolution of the Poisson bracket

We would like to understand the constraints on the functions $A, B, D_\pm$ coming from the identity

$$\frac{d}{dt}\{u, \tilde{u}\} = \left\{\frac{du}{dt}, \tilde{u}\right\} - \left\{\frac{d\tilde{u}}{dt}, u\right\}, \tag{A.5}$$

together with its $\{u, \tilde{v}\}$ and $\{v, \tilde{v}\}$ counterparts, where as before $\frac{df}{dt} = \dot{f} - \{H, f\}$ and $\dot{f} \equiv \partial_t f$. Since $\dot{u} = 1$ and otherwise only the unknown function $D_-$ could in principle explicitly depend on time, the above equation reduces to

$$2\mu^2\dot{D}_- - 2\mu^2\dot{\tilde{D}}_- + \{H, \{u, \tilde{u}\}\} = \{\{H, u\}, \tilde{u}\} - \{\{H, \tilde{u}\}, u\}. \tag{A.6}$$

We subsequently plug in the explicit Poisson brackets (2.31) and

$$\{H, u\} = \frac{\mu(\mathcal{H} - \mathcal{P})}{1 + 2\mu\mathcal{H}}(1 - 2\mu\mathcal{P}) \equiv \mu\Gamma_-, \tag{A.7}$$

which follows from (A.3), after having imposed charge conservation to cancel the winding contributions. Furthermore,

$$\{\mathcal{H}, \tilde{u}\} = -\mu\left[(G_-' + \mu F')\Theta(\tilde{\sigma} - \sigma) - 2G_-\delta(\sigma - \tilde{\sigma}) + D_-' + \mu B'\right], \tag{A.8}$$

$$\{\mathcal{P}, \tilde{u}\} = \mu\left[(G_-' - \mu F')\Theta(\tilde{\sigma} - \sigma) - 2G_-\delta(\sigma - \tilde{\sigma}) + D_-' - \mu B'\right], \tag{A.9}$$

obtained by taking a derivative of (2.31). It is worth noting the identities

$$G_\pm + \mu F = \mathcal{H} \pm \mathcal{P}, \quad G_\pm - \mu F = \frac{(\mathcal{H} \pm \mathcal{P})(1 \pm 2\mu\mathcal{P})}{1 + 2\mu\mathcal{H}} \equiv \Gamma_\pm. \tag{A.10}$$

For consistency, the terms proportional to $\Theta(\tilde{\sigma} - \sigma)$ and $\delta(\tilde{\sigma} - \sigma)$ in (A.6) should cancel separately, and they do. We are then left with terms that either depend only on $\sigma$, or only on $\tilde{\sigma}$. Thus we find two equations that must be satisfied separately,

$$2\{H, D_-\} - 2\dot{D}_- = \partial_\mathcal{H}\Gamma_-(D_- + \mu B)' - \partial_\mathcal{P}\Gamma_-(D_- - \mu B)', \tag{A.11}$$

as well as the same equation with $\sigma \to \tilde{\sigma}$, where

$$\partial_\mathcal{H}\Gamma_- = \frac{1 - 4\mu^2\mathcal{P}^2}{(1 + 2\mu\mathcal{H})^2}, \quad \partial_\mathcal{P}\Gamma_- = \frac{4\mu\mathcal{P}}{1 + 2\mu\mathcal{H}} - 1. \tag{A.12}$$

Next, it is useful to parametrize $D_-$ and $B$ as

$$D_- = G_-(\bar{X} - 1/2) - \mu^2 F X, \quad B = F(\bar{X} - 1/2) - G_+ X, \tag{A.13}$$

case in which (A.11) reduces to the following constraints on $X, \bar{X}$

$$\dot{X} - \{H, X\} = \frac{1 + 2\mu\mathcal{P}}{1 + 2\mu\mathcal{H}} X', \qquad \dot{\bar{X}} - \{H, \bar{X}\} = -\frac{1 - 2\mu\mathcal{P}}{1 + 2\mu\mathcal{H}} \bar{X}'. \tag{A.14}$$

Remembering that the field-dependent coordinates satisfy (2.12)

$$\frac{du}{dt} = \frac{1 - \mu\bar{\mathcal{L}}}{1 + \mu\mathcal{L}} u' = \frac{1 + 2\mu\mathcal{P}}{1 + 2\mu\mathcal{H}} u' = 1 - \{H, u\}, \qquad \frac{dv}{dt} = -\frac{1 - \mu\mathcal{L}}{1 + \mu\bar{\mathcal{L}}} v' = -\frac{1 - 2\mu\mathcal{P}}{1 + 2\mu\mathcal{H}} v' = -1 - \{H, v\}, \tag{A.15}$$

it is clear that $X$ should only be a function of $u$ and $\bar{X}$, only a function of $v$.

An identical argument for the $\{v, \tilde{v}\}$ bracket fixes the functions $A$ and $D_+$ to take the form (2.35), where we used again charge conservation to reduce the $\{H, v\}$ bracket to

$$\{H, v\} = -\frac{\mu(\mathcal{H} + \mathcal{P})}{1 + 2\mu\mathcal{H}}(1 + 2\mu\mathcal{P}) \equiv -\mu\Gamma_+. \tag{A.16}$$

Finally, the time derivative of the mixed Poisson bracket is

$$\frac{d}{dt}\{u, \tilde{v}\} = \left\{\frac{du}{dt}, \tilde{v}\right\} - \left\{\frac{d\tilde{v}}{dt}, u\right\}. \tag{A.17}$$

It is easy to see all terms proportional to $\Theta$ and $\delta$ functions cancel. We are left with

$$2\mu(\dot{A} - \{H, A\}) + (D_+ + \mu A)' \partial_{\mathcal{H}} \Gamma_- + (D_+ - \mu A)' \partial_{\mathcal{P}} \Gamma_- = 0 \tag{A.18}$$

$$2\mu(\dot{\tilde{B}} - \{H, \tilde{B}\}) - (\tilde{D}_- + \mu\tilde{B})' \partial_{\tilde{\mathcal{H}}} \tilde{\Gamma}_+ + (\tilde{D}_- - \mu\tilde{B})' \partial_{\tilde{\mathcal{P}}} \tilde{\Gamma}_+ = 0. \tag{A.19}$$

Interestingly, the $\{u, v\}$ Poisson bracket does not link the ambiguities on the left-moving side (parametrized by $B, D_-$, or $X, \bar{X}$) to those on the right-moving side (parametrized by $A, D_+$, or $Y, \bar{Y}$), as we find that the general solution (2.35) satisfies identically these equations for $X, Y$ holomorphic and $\bar{X}, \bar{Y}$ anti-holomorphic.

## A.3. Poisson brackets of the charges

In this subsection, we show explicitly the details of the calculation of the Poisson bracket between a left-moving charge $Q_f$ and a right-moving charge $\bar{Q}_{\tilde{f}}$. Plugging in the expressions (2.24), we find

$$
\begin{aligned}
\{Q_f, \bar{Q}_{\tilde{f}}\} = \frac{1}{4} \int d\sigma d\tilde{\sigma} \Big[ & f\tilde{\bar{f}}\{\mathcal{H} + \mathcal{P}, \tilde{\mathcal{H}} - \tilde{\mathcal{P}}\} + f'(\mathcal{H} + \mathcal{P})\tilde{\bar{f}}\{\hat{u}, \tilde{\mathcal{H}} - \tilde{\mathcal{P}}\} \\
& + f\tilde{\bar{f}}'(\tilde{\mathcal{H}} - \tilde{\mathcal{P}})\{\mathcal{H} + \mathcal{P}, \hat{\tilde{v}}\} + f'(\mathcal{H} + \mathcal{P})\tilde{\bar{f}}'(\tilde{\mathcal{H}} - \tilde{\mathcal{P}})\{\hat{u}, \hat{\tilde{v}}\} \Big] \\
= \frac{\mu}{2} \int d\sigma d\tilde{\sigma} \Big[ & -f\tilde{\bar{f}} F' \delta(\sigma - \tilde{\sigma}) - f' \frac{\mathcal{H} + \mathcal{P}}{R_u} \tilde{\bar{f}} \left(2\tilde{G}_- \delta(\sigma - \tilde{\sigma}) - \tilde{G}_-' \Theta(\sigma - \tilde{\sigma}) - \tilde{D}_-' + \hat{u}\tilde{G}_-'\right) \\
& + f\tilde{\bar{f}}' \frac{\tilde{\mathcal{H}} - \tilde{\mathcal{P}}}{R_v} \left(-2G_+ \delta(\sigma - \tilde{\sigma}) + G_+' \Theta(\tilde{\sigma} - \sigma) + D_+' - \hat{\tilde{v}}G_+'\right) \\
& + \mu^2 f' \frac{\mathcal{H} + \mathcal{P}}{R_u} \tilde{\bar{f}}' \frac{\tilde{\mathcal{H}} - \tilde{\mathcal{P}}}{R_v} \Big(F\Theta(\tilde{\sigma} - \sigma) + \tilde{F}\Theta(\sigma - \tilde{\sigma}) + A(\sigma) + B(\tilde{\sigma}) \\
& \qquad\qquad - \hat{u}(\tilde{F} - F(0) + w_A) - \hat{\tilde{v}}(F - F(0) + w_B)\Big) \Big].
\end{aligned}
\tag{A.20}
$$

We integrate by parts all terms that contain $F'$, $G'_\pm$ or $D'_\pm$, keeping in mind that the boundary terms do not vanish if the functions have winding. For example

$$\frac{\mu}{2}\int d\tilde{\sigma}\,\bar{\tilde{f}}\left(\tilde{G}'_-\Theta(\sigma-\tilde{\sigma})+\tilde{D}'_--\hat{u}\tilde{G}'_-\right) \tag{A.21}$$

$$=-\frac{\mu}{2}\bar{f}(0)(G_-(0)-w_{D_-})-\frac{\mu}{2}\int d\sigma\left[\bar{\tilde{f}}'\hat{v}'\left(\tilde{G}_-\Theta(\sigma-\tilde{\sigma})+\tilde{D}_--\hat{u}\tilde{G}_-\right)-\bar{\tilde{f}}\tilde{G}_-\delta(\sigma-\tilde{\sigma})\right].$$

Collecting furthermore the terms proportional to $\delta$ functions, we obtain

$$\{Q_f,\bar{Q}_{\tilde{f}}\}=\frac{\mu}{2}\int d\sigma\left[f'\bar{\tilde{f}}\left(\hat{u}'F-\frac{\mathcal{H}+\mathcal{P}}{R_u}G_-\right)+f\bar{\tilde{f}}'\left(\hat{v}'F-\frac{\mathcal{H}-\mathcal{P}}{R_v}G_+\right)\right] \tag{A.22}$$

$$+\frac{\mu}{2}\int d\sigma d\tilde{\sigma}\left[-f'\frac{\mathcal{H}+\mathcal{P}}{R_u}\bar{\tilde{f}}'\hat{v}'\left(\tilde{G}_-\Theta(\sigma-\tilde{\sigma})+\tilde{D}_--\hat{u}\tilde{G}_-\right)\right.$$

$$-f'\hat{u}'\bar{\tilde{f}}'\frac{\tilde{\mathcal{H}}-\tilde{\mathcal{P}}}{R_v}\left(G_+\Theta(\tilde{\sigma}-\sigma)+D_+-\hat{v}G_+\right)$$

$$+\mu^2f'\frac{\mathcal{H}+\mathcal{P}}{R_u}\bar{\tilde{f}}'\frac{\tilde{\mathcal{H}}-\tilde{\mathcal{P}}}{R_v}\left(F\Theta(\tilde{\sigma}-\sigma)+\tilde{F}\Theta(\sigma-\tilde{\sigma})+A(\sigma)+B(\tilde{\sigma})\right.$$

$$\left.\left.-\hat{u}(\tilde{F}-F(0)+w_A)-\hat{v}(F-F(0)+w_B)\right)\right]$$

$$-\frac{\mu}{R_u}Q_{f'}\bar{f}(0)(G_-(0)-w_{D_-})-\frac{\mu}{R_v}\bar{Q}_{\tilde{f}'}f(0)(G_+(0)-w_{D_+}). \tag{A.23}$$

The first line vanishes identically, using (2.25). The terms proportional to $\Theta$ functions add up to

$$\frac{\mu}{2}\int d\sigma d\tilde{\sigma}\left[f'\frac{\mathcal{H}+\mathcal{P}}{R_u}\frac{\bar{\tilde{f}}'}{R_v}\left(-\tilde{v}'\tilde{G}_-+\mu^2(\tilde{\mathcal{H}}-\tilde{\mathcal{P}})\tilde{F}\right)\Theta(\sigma-\tilde{\sigma})\right.$$

$$\left.+\frac{f'}{R_u}\bar{\tilde{f}}'\frac{\tilde{\mathcal{H}}-\tilde{\mathcal{P}}}{R_v}\left(-u'G_++\mu^2(\mathcal{H}+\mathcal{P})F\right)\Theta(\tilde{\sigma}-\sigma)\right]$$

$$=-\frac{\mu}{2}\int d\sigma d\tilde{\sigma}f'\bar{\tilde{f}}'\frac{\mathcal{H}+\mathcal{P}}{R_u}\frac{\tilde{\mathcal{H}}+\tilde{\mathcal{P}}}{R_v}[\Theta(\sigma-\tilde{\sigma})+\Theta(\tilde{\sigma}-\sigma)]=-\frac{2\mu}{R_uR_v}Q_{f'}\bar{Q}_{\tilde{f}'}. \tag{A.24}$$

Similarly, the terms proportional to $\hat{u}$ and $\hat{v}$ give

$$\frac{\mu}{R_u^2}Q_{uf'}\int d\tilde{\sigma}\frac{\bar{\tilde{f}}'}{R_v}\left[\tilde{v}'\tilde{G}_--\mu^2(\tilde{\mathcal{H}}-\tilde{\mathcal{P}})(\tilde{F}-F(0)+w_A)\right]$$

$$+\frac{\mu}{R_v^2}\bar{Q}_{v\tilde{f}'}\int d\sigma\frac{f'}{R_u}\left[u'G_+-\mu^2(\mathcal{H}+\mathcal{P})(F-F(0)+w_B)\right]$$

$$=\frac{2\mu}{R_uR_v}\left[\frac{1}{R_u}Q_{uf'}\bar{Q}_{\tilde{f}'}(1+\mu^2[F(0)-w_A])+\frac{1}{R_v}Q_{f'}\bar{Q}_{v\tilde{f}'}(1+\mu^2[F(0)-w_B])\right]. \tag{A.25}$$

Using these equations to work out the Poisson bracket of the charges, we finally find

$$\{Q_f, \bar{Q}_{\bar{f}}\} \tag{A.26}$$

$$= \frac{\mu}{R_u} Q_{f'} \Bigg( \bar{f}(0)(w_{D_-} - G_-(0))$$

$$+ \int d\tilde{\sigma} \frac{\tilde{\bar{f}}'}{R_v} \bigg[ (\tilde{\mathcal{H}} - \tilde{\mathcal{P}}) \Big( -\frac{1}{2} + \hat{\tilde{v}}(1 + \mu^2[F(0) - w_B]) + \mu^2 \tilde{B} \Big) - \tilde{v}' \tilde{D}_- \bigg] \Bigg)$$

$$+ \frac{\mu}{R_v} \bar{Q}_{\bar{f}'} \Bigg( f(0)(w_{D_+} - G_+(0))$$

$$+ \int d\sigma \frac{f'}{R_u} \bigg[ (\mathcal{H} + \mathcal{P}) \Big( -\frac{1}{2} + \hat{u}(1 + \mu^2[F(0) - w_A]) + \mu^2 A \Big) - u' D_+ \bigg] \Bigg).$$

The terms proportional to $u, v$ in the integrand, which represent the contribution of the state-dependent radii to the Poisson brackets, are dangerous, as they would produce terms proportional to $Q_{uf'}$ on the right-hand side, which are not conserved due to the non-periodicity of the associated function. A very similar derivation for the left-left and right-right components of the Poisson bracket algebra gives

$$\{Q_f, Q_g\} = \frac{1}{R_u} Q_{fg'-f'g}$$

$$- \frac{\mu^2}{R_u} Q_{f'} \Bigg( g(0)(F(0) - w_B) + \int \frac{d\tilde{\sigma}}{R_u} \tilde{g}' \big[ \tilde{u}' \tilde{B} - (\tilde{\mathcal{H}} + \tilde{\mathcal{P}}) \big( \tilde{D}_- + \hat{\tilde{u}}[G_-(0) - w_{D_-}] \big) \big] \Bigg)$$

$$+ \frac{\mu^2}{R_u} Q_{g'} \Bigg( f(0)(F(0) - w_B) + \int \frac{d\sigma}{R_u} f' \big[ u'B - (\mathcal{H} + \mathcal{P}) \big( D_- + \hat{u}[G_-(0) - w_{D_-}] \big) \big] \Bigg). \tag{A.27}$$

$$\{\bar{Q}_{\bar{f}}, \bar{Q}_{\bar{g}}\} = \frac{1}{R_v} \bar{Q}_{\bar{f}'\bar{g}-\bar{f}\bar{g}'}$$

$$- \frac{\mu^2}{R_v} \bar{Q}_{\bar{f}'} \Bigg( \bar{g}(0)(F(0) - w_A) + \int \frac{d\tilde{\sigma}}{R_v} \tilde{\bar{g}}' \big[ \tilde{v}' \tilde{A} - (\tilde{\mathcal{H}} - \tilde{\mathcal{P}}) \big( \tilde{D}_+ + \hat{\tilde{v}}[G_+(0) - w_{D_+}] \big) \big] \Bigg)$$

$$+ \frac{\mu^2}{R_v} \bar{Q}_{\bar{g}'} \Bigg( \bar{f}(0)(F(0) - w_A) + \int \frac{d\sigma}{R_v} \bar{f}' \big[ v'A - (\mathcal{H} - \mathcal{P}) \big( D_+ + \hat{v}[G_+(0) - w_{D_+}] \big) \big] \Bigg). \tag{A.28}$$

With these results, we can deduce how to choose the functions $A$, $B$ and $D_\pm$ in order to guarantee, for consistency, that the right-hand sides are conserved. In the following, we will analyse this requirement step by step.

Concretely, we need to cancel the radius contributions proportional with $Q_{uf'}$ and $\bar{Q}_{v\bar{f}'}$, which are not conserved due to the lack of periodicity of the functions that label them. Let us first consider the result (A.27), and understand the possible choices for the winding of $D_-$ so as to cancel this term. There are three types of winding one can consider: proportional to $\hat{u}(\sigma)$, proportional to $\Theta(\sigma)$, or proportional to $\hat{v}(\sigma)$. In the case of $D_-$, the first type of winding does not help cancel the non-conserved terms proportional to $Q_{uf'}$ in the total Poisson bracket. Moreover, it introduces $u$-type winding in (A.26), which cannot be cancelled by $u$-type winding in $B$ (or the other functions) without introducing additional $u$-type winding in (A.27). Consequently, the $u$-type winding in $B$ and $D_-$ must be zero. A similar argument sets the $v$-type winding in $A$ and $D_+$ to zero.

For whatever the other type of winding we may have, the vanishing of the terms proportional to $Q_{uf'}$ in (A.27) and $\bar{Q}_{v\bar{f}'}$ in (A.28) sets $w_{D_\pm} = G_\pm(0)$. The conservation equation (A.4)

then requires that $w_A = w_B = F(0)$, so the charge algebra becomes

$$\{Q_f, \bar{Q}_{\bar{f}}\} = \frac{\mu Q_{f'}}{R_u R_v} \int d\sigma \bar{f}' \left[ \left( \hat{v} - \frac{1}{2} + \mu^2 B \right)(\mathcal{H} - \mathcal{P}) - v' D_- \right]$$

$$+ \frac{\mu \bar{Q}_{\bar{f}'}}{R_u R_v} \int d\sigma f' \left[ \left( \hat{u} - \frac{1}{2} + \mu^2 A \right)(\mathcal{H} + \mathcal{P}) - u' D_+ \right], \qquad (A.29)$$

$$\{Q_f, Q_g\} = \frac{1}{R_u} Q_{fg'-f'g} - \frac{\mu^2}{R_u^2} Q_{f'} \int d\sigma g'(u'B - (\mathcal{H} + \mathcal{P})D_-)$$

$$+ \frac{\mu^2}{R_u^2} Q_{g'} \int d\sigma f'(u'B - (\mathcal{H} + \mathcal{P})D_-)$$

and a similar equation for $\{\bar{Q}_{\bar{f}}, \bar{Q}_{\bar{g}}\}$. The first term in the first equation can be cancelled if we choose a $v$ - type of winding $D_-$ and/or $B$; however, if we do not want it to introduce non-conserved terms in the second equation, we need to choose

$$u'B^{(v)} - (\mathcal{H} + \mathcal{P})D_-^{(v)} = 0, \qquad v'D_-^{(v)} - \mu^2(\mathcal{H} - \mathcal{P})B^{(v)} = \mathcal{H} - \mathcal{P}, \qquad (A.30)$$

where the superscript $(v)$ indicates the coefficient of $\hat{v}$ in the corresponding function. These equations determine $D_-^{(v)} = G_-$ and $B^{(v)} = F$. Since the total winding of the functions $D_-$ and $B$ is $G_-(0)$ and respectively $F(0)$, we see that the $v$-type winding accounts for all of it, leaving no room for a $\Theta$ - type of winding. An identical argument sets

$$v'A^{(u)} - (\mathcal{H} - \mathcal{P})D_+^{(u)} = 0, \qquad u'D_+^{(u)} - \mu^2(\mathcal{H} + \mathcal{P})A^{(u)} = \mathcal{H} + \mathcal{P}. \qquad (A.31)$$

Therefore, we find that the winding part of the functions $A, B, D_\pm$ is entirely fixed by charge conservation and the non-appearance of non-conserved terms in the charge algebra.

This however is not sufficient to fix the charge algebra, as the periodic parts of these functions have not yet been fixed. The integrals we are left with are only conserved if they are of the form $\mathcal{H} + \mathcal{P}$ times a periodic function of $\hat{u}$ or $\mathcal{H} - \mathcal{P}$ times a function of $\hat{v}$. Combining the first line of eq. (A.29) with the third, we find the requirement

$$(\tilde{\mathcal{H}} - \tilde{\mathcal{P}})\left( -\frac{1}{2} + \mu^2 \tilde{B}^{(p)} \right) - \tilde{v}'\tilde{D}_- = -(\tilde{\mathcal{H}} - \tilde{\mathcal{P}})\bar{X}_p(\hat{v}),$$

$$\tilde{u}'\tilde{B}^{(p)} - (\tilde{\mathcal{H}} + \tilde{\mathcal{P}})\tilde{D}_-^{(p)} = -(\tilde{\mathcal{H}} + \tilde{\mathcal{P}})X_p(\hat{u}), \qquad (A.32)$$

where the superscript $(p)$ indicates the periodic part and where $X_p(\hat{u})$ and $\bar{X}_p(\hat{v})$ are periodic functions that are arbitrary for now. Comparing this with (A.13) however, we find that $X_p = X$ and $\bar{X}_p = \bar{X} - \hat{v}$.

Similarly, the other contributions to (A.26)–(A.28) are conserved when (2.35) holds with $Y_p \equiv Y - \hat{u}$ and $\bar{Y}_p \equiv \bar{Y}$ periodic. The resulting algebra is

$$\{Q_f, \bar{Q}_{\bar{f}}\} = -\frac{\mu}{R_u R_v}\left( Q_{f'}Q_{\bar{X}_p \bar{f}'} + \bar{Q}_{\bar{f}'}Q_{Y_p f'} \right),$$

$$\{Q_f, Q_g\} = \frac{1}{R_u}Q_{fg'-f'g} + \frac{\mu^2}{R_u^2}\left( Q_{f'}Q_{X_p g'} - Q_{g'}Q_{X_p f'} \right),$$

$$\{\bar{Q}_{\bar{f}}, \bar{Q}_{\bar{g}}\} = \frac{1}{R_v}\bar{Q}_{\bar{f}'\bar{g}-\bar{f}\bar{g}'} + \frac{\mu^2}{R_v^2}\left( \bar{Q}_{\bar{f}'}\bar{Q}_{\bar{Y}_p \bar{g}'} - \bar{Q}_{\bar{g}'}\bar{Q}_{\bar{Y}_p \bar{f}'} \right). \qquad (A.33)$$

We can finally rewrite $\partial_u f = f'/R_u$ and $\partial_v \bar{f} = \bar{f}'/R_v$ to recover eq. (2.39).

The undetermined functions $X_p, \bar{X}_p, Y_p, \bar{Y}_p$ could more generally be constrained by the Jacobi identities that the Poisson bracket algebra must satisfy. These involve product of $\delta$ function distributions, so their full analysis is rather involved. We will not pursue it here.[16]

---

[16]Note added in v2: to simplify the analysis, it is possible to study the Jacobi identities involving a "smeared"

# B. The $J\bar{T}$ charge algebra

In this appendix, we present some details of the calculation of Poisson brackets of the conserved charges in $J\bar{T}$ - deformed CFTs. As explained in the main text, the undeformed commutators (3.29) and the general formula (3.25) allow us to compute the commutator of the deformed right-moving Hamiltonian with various quantities of interest. These read

$$\{\mathcal{H}_R(\sigma), \mathcal{H}_R(\tilde{\sigma})\} = \frac{-\mathcal{H}_R^{(0)}(\sigma) - \mathcal{H}_R^{(0)}(\tilde{\sigma}) + \frac{\lambda^2}{2}\mathcal{H}_R(\sigma)\mathcal{H}_R(\tilde{\sigma})}{\sqrt{(1-\lambda\mathcal{J}_+(\sigma))^2 - \lambda^2\mathcal{H}_R^{(0)}(\sigma)}\sqrt{(1-\lambda\mathcal{J}_+(\tilde{\sigma}))^2 - \lambda^2\mathcal{H}_R^{(0)}(\tilde{\sigma})}}\partial_\sigma\delta(\sigma - \tilde{\sigma}),$$
(B.1)

$$\{\mathcal{P}(\sigma), \mathcal{H}_R(\tilde{\sigma})\} = \frac{\mathcal{H}_R^{(0)}(\sigma) + \mathcal{H}_R^{(0)}(\tilde{\sigma}) + \lambda\mathcal{J}_+(\sigma)\mathcal{H}_R(\tilde{\sigma})}{\sqrt{(1-\lambda\mathcal{J}_+(\tilde{\sigma}))^2 - \lambda^2\mathcal{H}_R^{(0)}(\tilde{\sigma})}}\partial_\sigma\delta(\sigma - \tilde{\sigma}),$$
(B.2)

$$\{\mathcal{H}_R(\sigma), \mathcal{J}_+(\tilde{\sigma})\} = \frac{\lambda\mathcal{H}_R(\sigma)}{2\sqrt{(1-\lambda\mathcal{J}_+(\sigma))^2 - \lambda^2\mathcal{H}_R^{(0)}}}\partial_\sigma\delta(\sigma - \tilde{\sigma}),$$
(B.3)

$$\{\mathcal{H}_R(\sigma), \mathcal{J}_-(\tilde{\sigma})\} = -\frac{\mathcal{J}_-(\sigma)}{\sqrt{(1-\lambda\mathcal{J}_+(\sigma))^2 - \lambda^2\mathcal{H}_R^{(0)}(\sigma)}}\partial_\sigma\delta(\sigma - \tilde{\sigma}).$$
(B.4)

Using the fact that $\mathcal{J}_+ - \mathcal{J}_- = \phi_1'$ and that the last two commutators are total $\tilde{\sigma}$ derivatives, we can deduce the commutator of $\mathcal{H}_R$ with $\phi$, which will be used to compute the Poisson bracket of the energy currents with the field-dependent coordinate $v = V - \lambda\phi_1$. We find

$$\{\mathcal{H}_R(\sigma), \phi_1(\tilde{\sigma})\} = \frac{-\mathcal{J}_-(\sigma) - \lambda\mathcal{H}_R(\sigma)/2}{\sqrt{(1-\lambda\mathcal{J}_+(\sigma))^2 - \lambda^2\mathcal{H}_R^{(0)}(\sigma)}}\delta(\sigma - \tilde{\sigma}).$$
(B.5)

Note that, in principle, we could have added an arbitrary integration function, $A(\sigma)$, to the right-hand-side of this commutator. However, the fact that the commutator is local with respect to $\sigma - \tilde{\sigma}$ compels us to set this potential integration function to zero.

Using the above commutators, we can easily compute those of $\mathcal{H}_L = \mathcal{H}_R + \mathcal{P}$. As noted in the main text, the $\{\mathcal{H}_L, \mathcal{H}_L\}$ commutator can be shown to be equivalent (using the criteria in appendix C) to

$$\{\mathcal{H}_L(\sigma), \mathcal{H}_L(\tilde{\sigma})\} = (\mathcal{H}_L(\sigma) + \mathcal{H}_L(\tilde{\sigma}))\partial_\sigma\delta(\sigma - \tilde{\sigma}).$$
(B.6)

The $\{\mathcal{H}_L(\sigma), \mathcal{H}_R(\tilde{\sigma})\}$ commutator is somewhat involved; however, in the calculations below, in which we will integrate over $\tilde{\sigma}$, it suffices to know that it is proportional to $\partial_\sigma\delta(\sigma - \tilde{\sigma})$ which implies, using (C.1), that we only need to know its value at $\tilde{\sigma} = \sigma$, as well as that of its first $\tilde{\sigma}$ derivative evaluated at $\tilde{\sigma} = \sigma$,

$$\{\mathcal{H}_L(\sigma), \mathcal{H}_R(\tilde{\sigma})\}|_{\tilde{\sigma}=\sigma} = \mathcal{H}_R\left(1 - \frac{1}{\sqrt{(1-\lambda\mathcal{J}_+)^2 - \lambda^2\mathcal{H}_R^{(0)}}}\right), \quad \partial_{\tilde{\sigma}}\{\mathcal{H}_L(\sigma), \mathcal{H}_R(\tilde{\sigma})\}|_{\tilde{\sigma}=\sigma} = 0.$$
(B.7)

---

version of the field dependent coordinates $u_0 \equiv \int_0^R d\sigma\, u(\sigma)$, $v_0 \equiv \int_0^R d\sigma\, v(\sigma)$ and the currents, as was done in [72] for $J\bar{T}$. The simplest Jacobi identity is $\{\{P, \tilde{P}\}, u_0\}$ + cyclic = 0. Plugging in the minimal solution, we find that this Jacobi identity *is not* satisfied. More generally, it seems to be very difficult to find a solution to the Jacobi identities that has the non-trivial winding required by the charge conservation equation (A.4). If no such solution exists, this means that the functions $u, v$ as defined in this note are not good functions on phase space when the $T\bar{T}$ - deformed theory lives on compact space. This does not, however, preclude the existence of related field-dependent coordinates whose action phase space is well-defined, as was the case for $J\bar{T}$.

The $\{\mathcal{H}_L, \phi\}$ commutator can be inferred from (B.5).

We can now compute the algebra of the symmetry generators (3.12). It is trivial to show that the left-movers satisfy a Virasoro algebra, since

$$\{Q_f, Q_g\} = \int d\sigma d\tilde{\sigma} f(U)g(\tilde{U})\{\mathcal{H}_L(\sigma), \mathcal{H}_L(\tilde{\sigma})\} = \int d\sigma \mathcal{H}_L(\sigma)(fg' - f'g) = Q_{fg'-f'g}. \quad \text{(B.8)}$$

The commutator of the left and the right generators can be shown to vanish

$$\{Q_f, \bar{Q}_{\bar{f}}\} = \int d\sigma d\tilde{\sigma} f(U)\big[\{\mathcal{H}_L(\sigma), \mathcal{H}_R(\tilde{\sigma})\}\bar{f}(\tilde{v}) + \bar{f}'(\tilde{v})\{\mathcal{H}_L(\sigma), \tilde{v}\}\mathcal{H}_R(\tilde{\sigma})\big] \quad \text{(B.9)}$$

$$= \int d\sigma f(U)\big[\{\mathcal{H}_L(\sigma), \mathcal{H}_R(\tilde{\sigma})\}|_{\tilde{\sigma}=\sigma} \partial_\sigma \bar{f}(v) - \lambda \bar{f}'(v)\{\mathcal{H}_L(\sigma), \phi(\tilde{\sigma})\}|_{\tilde{\sigma}=\sigma} \mathcal{H}_R(\sigma)\big] = 0,$$

where we used the fact that the field-dependent coordinate $v = V - \lambda\phi_1$ (and so $\partial_\sigma \bar{f} = \bar{f}'(1 - \lambda\phi_1')$), as well as the commutators (B.7) and we used the formula (C.5) for the integral of a derivative of a delta function.

Finally, the right-moving generators have commutation relations

$$\{\bar{Q}_{\bar{f}}, \bar{Q}_{\bar{g}}\} = \int d\sigma d\tilde{\sigma} \big[\bar{f}(v)\bar{g}(\tilde{v})\{\mathcal{H}_R, \tilde{\mathcal{H}}_R\} + \bar{f}'(v)\{v, \mathcal{H}_R(\tilde{\sigma})\}\mathcal{H}_R \bar{g}(\tilde{v}) + \bar{f}(v)\bar{g}'(\tilde{v})\{\mathcal{H}_R, \tilde{v}\}\tilde{\mathcal{H}}_R\big]$$

$$= \int d\sigma \big[\bar{f}(v)\,\partial_{\tilde{\sigma}}\big(\bar{g}(\tilde{v})\{\mathcal{H}_R, \tilde{\mathcal{H}}_R\}\big)\big|_{\tilde{\sigma}=\sigma} + \lambda(\bar{f}'\bar{g} - \bar{g}'\bar{f})\mathcal{H}_R\,\{\mathcal{H}_R(\sigma), \phi(\tilde{\sigma})\}|_{\tilde{\sigma}=\sigma}\big]. \quad \text{(B.10)}$$

Note there is no $\{v, \tilde{v}\}$ commutator, as $v = V - \lambda\phi_1$ only involves $\phi$. Using the fact that

$$\partial_{\tilde{\sigma}}\big(\{\mathcal{H}_R, \tilde{\mathcal{H}}_R\}\big)\big|_{\tilde{\sigma}=\sigma} = \frac{2}{\lambda^2}\partial_\sigma\left(1 - \frac{1 - \lambda\mathcal{J}_+}{\sqrt{(1 - \lambda\mathcal{J}_+)^2 - \lambda^2\mathcal{H}_R^{(0)}}}\right) = -\partial_\sigma \frac{\mathcal{H}_R}{\sqrt{(1 - \lambda\mathcal{J}_+)^2 - \lambda^2\mathcal{H}_R^{(0)}}}. \quad \text{(B.11)}$$

Integrating by parts and using the specific values of the commutators, we find

$$\{\bar{Q}_{\bar{f}}, \bar{Q}_{\bar{g}}\} = \int d\sigma(\bar{f}'\bar{g} - \bar{g}'\bar{f})\mathcal{H}_R = \bar{Q}_{\bar{f}'\bar{g} - \bar{f}\bar{g}'}. \quad \text{(B.12)}$$

Let us now compute the Poisson brackets with the currents. It is easy to show, using the explicit expression (3.16) for $K_U$, that

$$\{Q_f, P_\chi\} = \int d\sigma d\tilde{\sigma} f(U)\chi(\tilde{U})\{\mathcal{H}_L(\sigma), K_U(\tilde{\sigma})\} = \int d\sigma f(U)\chi'(U)K_U(\sigma) = P_{f\chi'}. \quad \text{(B.13)}$$

The commutator of two $U(1)$ charges gives the usual central extension, $\{P_\chi, P_\eta\} = 1/2 \int d\sigma \chi\eta'$ and the commutator with the right-moving Virasoro generators vanishes. As for the right-moving $U(1)$

$$\bar{P}_{\bar{\chi}} = \int d\sigma \frac{\pi - \phi'}{2}\bar{\chi}(v), \quad \text{(B.14)}$$

it can be easily shown that they commute with the left-moving $U(1)$ and the left-moving Virasoro. The algebra of these $U(1)$ currents is however not Kac-Moody

$$\{\bar{P}_{\bar{\chi}}, \bar{P}_{\bar{\eta}}\} = \frac{1}{2}\int d\sigma\left(-\bar{\chi}\bar{\eta}' + \lambda\bar{\chi}\bar{\eta}'\frac{\pi + \phi'}{2} - \lambda\bar{\chi}'\bar{\eta}\frac{\pi - \phi'}{2}\right), \quad \text{(B.15)}$$

$$\{\bar{Q}_{\bar{f}}, \bar{P}_{\bar{\eta}}\} = -\int d\sigma \left( \bar{f} \bar{\eta}' \mathcal{J}_- + \frac{\lambda}{2} \bar{f}' \bar{\eta} \mathcal{H}_R \right). \tag{B.16}$$

As explained in the main text, we can obtain an operator that has the standard commutator with the right-moving Virasoro by adding a multiple of $T_{\alpha\bar{z}}$ to the current. It turns out that the combination

$$\bar{P}_{\bar{\eta}}^{KM} = \int d\sigma \left( \frac{\pi - \phi'}{2} + \frac{\lambda}{2} \mathcal{H}_R \right) \bar{\eta} \tag{B.17}$$

does the job. The commutator of these new charges is

$$\{\bar{P}_{\bar{\chi}}, \bar{P}_{\bar{\eta}}\} = -\frac{1}{2} \int d\sigma \, \bar{\chi} \bar{\eta}' (1 - \lambda\phi') = -\frac{1}{2} \int d\sigma \, \bar{\chi} \partial_\sigma \bar{\eta}. \tag{B.18}$$

Thus, we find two copies of the Virasoro - Kac-Moody algebra. As explained in the main text, on a compact space we need to divide the coordinates by the associated radii, but this does not affect the charge algebra, because the zero modes of $\mathcal{J}_\pm$ that enter $R_\nu$ commute with all the currents.

## C. Equivalence of distributions

The derivative of a Dirac-$\delta$ distribution acts on test functions as

$$\int d\tilde{\sigma} f(\sigma) g(\tilde{\sigma}) \partial_\sigma \delta(\sigma - \tilde{\sigma}) = f(\sigma) g'(\sigma). \tag{C.1}$$

It is useful to know when two distributions of the form $F(\sigma, \tilde{\sigma}) \partial_\sigma \delta(\sigma - \tilde{\sigma})$ are equivalent. The following criterion is necessary and sufficient:

$$F_1 \partial_\sigma \delta(\sigma - \tilde{\sigma}) \cong F_2 \partial_\sigma \delta(\sigma - \tilde{\sigma}) \Leftrightarrow \begin{cases} F_1(\sigma, \sigma) & = F_2(\sigma, \sigma) \\ \partial_\sigma F_1(\sigma, \tilde{\sigma})|_{\tilde{\sigma}=\sigma} & = \partial_\sigma F_2(\sigma, \tilde{\sigma})|_{\tilde{\sigma}=\sigma} \\ \partial_{\tilde{\sigma}} F_1(\sigma, \tilde{\sigma})|_{\tilde{\sigma}=\sigma} & = \partial_{\tilde{\sigma}} F_2(\sigma, \tilde{\sigma})|_{\tilde{\sigma}=\sigma} \end{cases} . \tag{C.2}$$

An example of equivalent distributions is given by $F_1 = \sqrt{f(\sigma)f(\tilde{\sigma})}$ and $F_2 = [f(\sigma) + f(\tilde{\sigma})]/2$, or

$$[f(\sigma)g(\tilde{\sigma}) + g(\sigma)f(\tilde{\sigma})] \partial_\sigma \delta(\sigma - \tilde{\sigma}) = [f(\sigma)g(\sigma) + f(\tilde{\sigma})g(\tilde{\sigma})] \partial_\sigma \delta(\sigma - \tilde{\sigma}), \tag{C.3}$$

which indeed satisfies eq. (C.2).

The remainder of this subsection is dedicated to showing eq. (C.2). To this end, note that two distributions $\mathcal{F}_1$ and $\mathcal{F}_2$ are equivalent iff they integrate to the same result against arbitrary test functions, i.e.

$$\int_{\sigma_1}^{\sigma_2} d\sigma \, \mathcal{F}_1 = f(\sigma_1, \sigma_2, \tilde{\sigma}) = \int_{\sigma_1}^{\sigma_2} d\sigma \, \mathcal{F}_2, \tag{C.4a}$$

$$\int_{\tilde{\sigma}_1}^{\tilde{\sigma}_2} d\tilde{\sigma} \, \mathcal{F}_1 = g(\tilde{\sigma}_1, \tilde{\sigma}_2, \sigma) = \int_{\tilde{\sigma}_1}^{\tilde{\sigma}_2} d\tilde{\sigma} \, \mathcal{F}_2, \tag{C.4b}$$

$$\int_{\sigma_1}^{\sigma_2} d\sigma \int_{\tilde{\sigma}_1}^{\tilde{\sigma}_2} d\tilde{\sigma} \, \mathcal{F}_1 = h(\sigma_1, \sigma_2, \tilde{\sigma}_1, \tilde{\sigma}_2) = \int_{\sigma_1}^{\sigma_2} d\sigma \int_{\tilde{\sigma}_1}^{\tilde{\sigma}_2} d\tilde{\sigma} \, \mathcal{F}_2. \tag{C.4c}$$

Obviously, if $f$ or $g$ are normal functions (i.e. not distributions), the requirement eq. (C.4c) follows from either eq. (C.4a) or eq. (C.4b). For distributions proportional to $\delta'$, the left-hand side of the requirement eq. (C.4a) gives

$$\int_{\sigma_1}^{\sigma_2} d\sigma \, F(\sigma, \tilde{\sigma}) \partial_\sigma \delta(\sigma - \tilde{\sigma}) \tag{C.5}$$
$$= F(\sigma_2, \sigma_2) \delta(\tilde{\sigma} - \sigma_2) - F(\sigma_1, \sigma_1) \delta(\tilde{\sigma} - \sigma_1) - \partial_\sigma F(\sigma, \tilde{\sigma})|_{\sigma = \tilde{\sigma}} \Theta(\sigma_1 < \tilde{\sigma} < \sigma_2).$$

The result is not a simple function, but it is a distribution with only $\delta$ functions for which we know the equivalence relations: the coefficients of the delta functions have to agree, as well as the remainder. This leads to the first two equations on the right-hand side of eq. (C.2). The third equation follows from eq. (C.4b). Equation (C.4c) does not provide additional constraints.

The fact that the result of eq. (C.5) contains $\delta$ functions, suggests a generalization to distributions of the form $[F(\sigma, \tilde{\sigma}) \partial_\sigma + G(\sigma, \tilde{\sigma})] \delta(\sigma - \tilde{\sigma})$. Two such distributions are equivalent iff

$$F_1(\sigma, \sigma) = F_2(\sigma, \sigma), \tag{C.6a}$$
$$\partial_\sigma F_1(\sigma, \tilde{\sigma}) - G_1(\sigma, \tilde{\sigma})|_{\sigma = \tilde{\sigma}} = \partial_\sigma F_2 - G_2(\sigma, \tilde{\sigma})|_{\sigma = \tilde{\sigma}}, \tag{C.6b}$$
$$\partial_{\tilde{\sigma}} F_1(\sigma, \tilde{\sigma}) + G_1(\sigma, \tilde{\sigma})|_{\sigma = \tilde{\sigma}} = \partial_{\tilde{\sigma}} F_2(\sigma, \tilde{\sigma}) + G_2(\sigma, \tilde{\sigma})|_{\sigma = \tilde{\sigma}}. \tag{C.6c}$$

The derivation is identical.

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
