# Peer review of "Infinite pseudo-conformal symmetries of classical $T \bar T$, $J \bar T $ and $J T_a$ - deformed CFTs"

_SciPost Physics, doi:SciPost Phys. 11, 078 (2021)_

## Round 1 · Referee Report · Anonymous (Referee 1) · 2020-12-30

Report

This paper continues the investigations concerning properties
of families of solvable, non-local, irrelevant deformations of two-dimensional CFTs -- $T\bar T$, $J\bar T$ and $JT_a$ --
which have, in particular, applications in various fields,
such as string theory, holography, black holes and QCD
-- concretely, their infinite symmetries, which are in one to one correspondence with the initial conformal and the affine $U(1)$ symmetries of the undeformed CFT.
It is an interesting work, which not only adds new ingredients to known facts, but also raises questions for future research,
and it is thus suitable for publication.
  • validity: -
  • significance: -
  • originality: -
  • clarity: -
  • formatting: -
  • grammar: -

Author:  Ruben Monten  on 2021-08-13  [id 1668]

(in reply to Report 1 on 2020-12-30)

We would like to thank the referee for their comments.

---

## Round 1 · Referee Report · Anonymous (Referee 2) · 2021-1-18

Report

This paper studies an infinite set of underlying symmetries in $T\bar{T}$, $J\bar{T}$, and $JT_a$ deformed conformal field theories (CFTs) in two-dimensions. In particular, the authors showed that the symmetries at the classical level are two commuting copies of a (functional) Wit algebra for the $T\bar{T}$-deformed CFTs and a (functional) Wit and a $U(1)$ Kac-Moody algebras for the $J\bar{T}$ and $JT_a$-deformed CFTs. In the case of deformed CFTs with large central charges, there exist gravity duals and these results corroborate the infinite symmetries previously found by the authors and a collaborator in the holographic setting.

As the authors emphasize, to some extent, the results are expected, without holography, from an alternative interpretation of the deformed theories; namely, a deformed theory can be thought of as the undeformed theory living on a field-dependent (or operator or state-dependent) deformed space. Their results match in detail with this expectation.

Technically, in order to discuss the precise algebras of these symmetries, the authors used and developed the Hamiltonian formalism of these deformed theories and computed the Poisson brakets of the symmetry generators (and the field-dependent coordinates). Not only do these analyses show useful detailed structures of the algebras, but they also reveal a subtle issue in the case of the $T\bar{T}$-deformed CFTs on the cylinder due to the field-dependent radius. The issue remains in the paper, but it gives useful data for the future and further studies.

I am slightly confused about the appearance of the Kac-Moody level in their classical analyses. If I am not mistaken, the Kac-Moody level is quantum in nature and furthermore, the central charge of the Virasoro algebra is related to the Kac-Moody level via Sugawara construction. I would then expect that especially in the free boson example, one would be able to find the Virasoro, rather than Wit, algebra by considering the Sugawara form of the Virasoro generators. This might well be simply my confusion and ignorance, but it would be better if the authors can comment on this point.

The paper is thorough and clearly written, and the authors made it easy to read. It clearly deserves a publication in SciPost and I thus recommend the paper for publication (hopefully with the above point on the KM level and the central charge being addressed).
  • validity: high
  • significance: high
  • originality: high
  • clarity: high
  • formatting: excellent
  • grammar: excellent

Author:  Ruben Monten  on 2021-08-13  [id 1667]

(in reply to Report 2 on 2021-01-18)
Category:
answer to question

We would like to thank the referee for their comments.

Concerning the level of the Kac–Moody algebra and its relation to the Virasoro central charge through the Sugawara construction, the following argument readily shows that there is a nonzero level already for classical bosonic fields, using the (undeformed) classical free boson as an example. Here, we can use the Poisson bracket algebra to find the classical analog of the Kac–Moody level. Using the classical analogue of the Sugawara construction, one can use this to recover the Witt terms in the stress tensor algebra.

More explicitly, we can use the free boson's chiral current $J^+ = \partial_U \phi \, d U$ together with the undeformed Poisson bracket $\{ \phi(x_1), \dot{\phi}(x_2) \} = \delta(x_1 - x_2)$ to find the Kac–Moody Poisson bracket algebra
\begin{align*}
\{ J^+(x_1), J^+(x_2) \} &= \frac12 \delta'(x_1 - x_2)
\ .
\end{align*}
Expanding the current in modes $J^+(U) = \sum_n e^{i n U} J_n^+$, we find
\begin{align*}
\{ J_m^+, J_n^+ \}
= \frac1{4\pi^2} \int dx_1 dx_2 e^{-i(m U_1 + n U_2)} \frac12 \delta'(x_1 - x_2)
= i \pi m \delta_{m+n}
\ ,
\end{align*}
which shows that the Kac–Moody algebra has a nonzero (although classically unquantized) level.

The left-moving stress tensor component $T \equiv T_{UU} = (\partial_U \phi)^2 = (J^+)^2$ is indeed “pure Sugawara”. Its Poisson bracket is
\begin{align*}
\{ T(x_1), T(x_2) \} = 2 J^+(x_1) J^+(x_2) \delta'(x_1 - x_2)
\ ,
\end{align*}
or in terms of the modes $T(U) = \sum_n e^{i n U} T_n$ we find
\begin{align*}
\{ T_m, T_n \}
&= \frac1{2\pi^2} \int d x \, e^{-i (m + n) U} \left( i m [J^+(x)]^2 - J^+(x) J^+{}'(x) \right)
\\
&= \frac{i}{2\pi} (m - n) T_{m+n}
\ .
\end{align*}
We thus find the Witt part of the Virasoro algebra. Note however that one cannot recover the central term in the Virasoro algebra, as this is related to the normal-ordering inherent in the quantum theory.

---

## Round 2 · Referee Report · Anonymous (Referee 2) · 2021-9-11

Report

I am happy to accept this article for publication. My question based on confusion has been clarified by the authors. I apologize to the authors for taking an unnecessarily long time to review the revision.

---

## Round 2 · Author Response

• Updated the discussion concerning the additional constraints on the undetermined functions in the TTbar-deformed charge algebra.
  • Corrected typos.

---

## Editorial Decision

published